

# How does the Environment Modulate Aerosol Impacts on Tropical Sea Breeze Convective Systems?

J. Minnie Park[1,2] and Susan C. van den Heever[1]

[1] Department of Atmospheric Science, Colorado State University, Fort Collins, Colorado

[2] Environmental and Climate Sciences Department, Brookhaven National Laboratory, Upton, New York

*Correspondence to*: J. Minnie Park (jpark1@bnl.gov)

**Abstract.**

This study investigates how the enhanced loading of microphysically and radiatively active aerosol particles impacts tropical sea breeze convection and whether these aerosol impacts are modulated by the multitudinous environments that

support these cloud systems. To achieve these goals, we have performed two large numerical model ensembles, each comprised of 130 idealised simulations that represent different initial conditions typical of tropical sea breeze environments. The two ensembles are identical with the exception of the fact that one ensemble is initialised with relatively low aerosol loading or pristine conditions, while the other is initialised with higher aerosol loading or polluted conditions. Six atmospheric and four surface parameters are simultaneously perturbed for the 130 initial conditions. Analysis of the ten-dimensional parameter

simulations was facilitated by the use of a statistical emulator and multivariate sensitivity techniques.

Comparisons of the clean and polluted ensembles demonstrate that aerosol direct effects reduce the incoming shortwave radiation reaching the surface, as well as the outgoing longwave radiation, within the polluted ensemble. This results in weaker surface fluxes, a reduced ocean-land thermal gradient, and a weaker sea breeze circulation. Consequently, irrespective of the different initial environmental conditions, increasing aerosol concentration decreases the three ingredients

necessary for moist convection: moisture, instability, and lift. As reduced surface fluxes and instability inhibit the convective boundary layer development, updraft velocities of the daytime cumulus convection developing ahead of the sea breeze front are robustly reduced in the polluted environments. Furthermore, the variance-based sensitivity analysis reveals that the soil saturation fraction is the most important environmental factor contributing to the updraft velocity variance of this daytime cumulus mode, but that it becomes a less important contributor with enhanced aerosol loading.

It is also demonstrated that enhanced aerosol loading results in a weakening of the convection initiated along the sea breeze front. This suppression is particularly robust when the sea breeze-initiated convection is shallower, and hence restricted to warm rain processes. However, when the sea breeze-initiated convection is deep and includes mixed-phase processes, both the sign and magnitude of the convective updraft responses to increased aerosol loading are modulated by the environment. The less favourable convective environment arising from aerosol direct effects also restricts the development of sea breeze-

initiated deep convection. While precipitation is ubiquitously suppressed with enhanced aerosol loading, the magnitude of this suppression remains a function of the initial environment. Altogether, our results highlight the importance of evaluating aerosol impacts on convection systems under the wide range of environments supporting such convective development.



# 1 Introduction

Sea breeze convective systems are one of the key contributors to coastal cloudiness and rainfall in the tropics (Keenan
and Carbone, 2008; Qian, 2008; Giangrande et al., 2014; Wang and Sobel, 2017). Differential heating over the land and ocean
induces thermally-driven baroclinic circulations, which modulate coastal air temperatures, impact coastal relative humidities,
and redistribute coastal aerosols (Miller et al., 2003). Such organised tropical convection also plays an essential role in global
climates via its impacts on planetary circulations such as the Walker circulation or the Madden-Julian Oscillation (Hendon and
Woodberry, 1993; Zhang, 2005). Given their importance, a number of efforts have been made to examine how different
atmospheric or land-surface parameters modify tropical sea breeze convective systems (Qian et al., 2012; Grant and van den
Heever, 2014; Bergemann and Jakob, 2016; Igel et al. 2018; Park et al., 2020), as well as to improve their representation in
numerical weather models (Boyle and Klein, 2010; Bergemann et al., 2017; Brown et al., 2017). However, in spite of these
past studies, accurately predicting sea breeze convective systems remains challenging (Kidd et al., 2013; Azorin-Molina et al.,
2014; Chen et al., 2015; Banta et al., 2020; Short, 2020). This is due in part to the large number of environmental parameters
that impact sea breeze convective system, and in part to the fact that these parameters coexist, covary, and interact with one
another (Crosman and Horel, 2010), which requires the use of sophisticated statistical approaches to identify those parameters
predominantly responsible for sea breeze convection (Igel et al., 2018; Park et al., 2020). Moreover, in association with the
continuous increase in coastal human populations, a rise in aerosol emissions due to anthropogenic activities and biomass
burning in the tropics has been observed (Reid et al., 2012; Wang et al., 2013; Menut et al., 2018). The presence of aerosols
can further complicate the behaviour of sea breeze convective system through radiative, microphysical, and dynamical
feedback processes. Aerosol emissions and processes may also, in turn, be modulated by the environmental parameters. While
the basic processes driving sea breeze convective systems are relatively well understood, the impacts of aerosols on sea breeze
convective systems, and the modulation of these impacts by various environmental properties, have not received much
attention.

Aerosol particles can interact directly with shortwave and longwave radiation, through scattering and absorption
(McCormick and Ludwig, 1967; Charlson and Pilat, 1969; Atwater, 1970; Mitchell Jr., 1971; Coakley Jr. et al., 1983). These
aerosol direct effects have numerous implications for processes important to the development of shallow and deep convection.
Scattering and absorption of shortwave and longwave radiation, and the subsequent reemission of longwave radiation at the
surface, influence surface sensible and latent heat fluxes and near-surface temperatures, all of which impact the development
of the convective boundary layer. A number of studies have highlighted the importance of considering both the radiative and
microphysical effects of aerosol particles on convective boundary layers and inversions. For example, within an aerosol–
radiation–land surface feedback framework, reduced surface sensible heat fluxes due to aerosol direct effects have been found
stabilize the lower troposphere, suppress boundary layer development and convectively available potential energy (CAPE),
and enhance the capping inversion (Yu et al., 2002). As a result, shallow convection and precipitation over land have been
found to become weaker (Jiang and Feingold, 2006; Niyogi et al., 2007). From these studies, it is clear that aerosol–radiation



interactions may have important feedbacks to the environment and the resulting sea breeze convective system. Nevertheless, studies of aerosol impacts on cloud properties, in particular deep convection, have quite often neglected aerosol–radiation interactions in the interest of focusing specifically on aerosol indirect effects (e.g., Storer and van den Heever, 2013; Miltenberger et al., 2018; Marinescu et al., 2021).

Depending on their size and composition, various aerosol particles can serve as cloud condensation nuclei (CCN). Under the assumption of the same liquid water content, when the number concentration of CCN is increased, a greater number of smaller cloud droplets are formed, thereby increasing the cloud albedo (Twomey, 1974) and the cloud lifetime (Albrecht, 1989), in what are often referred to as the first and second aerosol indirect effects, respectively. Less efficient collision and coalescence processes between the population of more numerous smaller cloud droplets suppresses the warm rain process,
resulting in the longer cloud lifetimes described in the second indirect effect. These aerosol effects were initially postulated for shallow convective clouds, but have also been observed to occur in deep convective cloud systems (Tao et al., 2012). It has been further hypothesised that with the suppression of the warm rain process, more numerous cloud droplets are lifted above the freezing level and frozen, thereby releasing additional latent heating and potentially strengthening updrafts. This process, which takes places in the mixed– and cold-phase regions of clouds, is often referred to as "cold-phase invigoration"
and has been reported in both observational and modelling studies (e.g., Andreae et al., 2004; Khain et al., 2005; Koren et al., 2005; van den Heever et al., 2006; Rosenfeld et al., 2008). Under increased aerosol loading, the condensational growth rate of the population of more numerous smaller cloud droplets within the warm phase regions of clouds has also been observed to increase due to the increased exposed total droplet surface area, thereby releasing more latent heating and enhancing convective updrafts closer to cloud base. These warm–phase aerosol–cloud dynamical feedbacks have collectively been termed
"condensational invigoration" (Kogan and Martin, 1994; Seiki and Nakajima, 2014; Saleeby et al., 2015; Sheffield et al., 2015) or "warm–phase invigoration" (Fan et al., 2018). It has recently been suggested that warm-phase invigoration may be more significant and robust than its cold-phase counterpart (Marinescu et al. 2021; Igel and van den Heever 2021).

        While convective invigoration theories have proposed stronger updrafts and/or heavier precipitation for convective clouds developing within enhanced aerosol loading conditions, a number of studies have questioned the robustness of these
theories for various reasons, one of which is the modulation of these effects by the cloud environment. For example, wind shear (Fan et al., 2009; Lebo and Morrison, 2014; Marinescu et al., 2017), CAPE (Lee et al., 2008; Storer et al., 2010; Storer et al. 2014), boundary layer instability (Marinescu et al., 2021) and moisture (Khain et al., 2005, 2008; Tao et al., 2007; Grant and van den Heever, 2015), have all been found to be modulating factors of aerosol impacts on cloud systems. Such environmental modulations have not been explored in the context of sea breeze convective systems, in spite of the fact that
they are one of the most ubiquitous forms of convection in the tropics (Hadi et al., 2002; Perez and Silva Dias., 2017), and as such, forms the focus of this paper. The primary goals of this study are twofold. The first goal is to investigate how radiatively and microphysically active aerosols change the convection that develops within tropical sea breeze regimes, specifically over the land. The second goal is to determine whether and how these convective responses to enhanced aerosol loading are modulated by the large number of environmental factors supporting sea breeze convective systems.





To achieve our stated goals, we have examined aerosol–cloud interactions within a fully interactive aerosol–
radiation–cloud–land surface framework and under a wide range of different tropical sea breeze environments. This is
accomplished through the use of a large ensemble of idealised numerical simulations in which we investigate the impact of
varying aerosol number concentrations on the characteristics of sea breeze convective systems developing under a wide range
of different initial environmental conditions previously identified in the literature as being important to such systems. More

specifically, we extend Park et al. (2020) in which we assessed the relative importance of ten different thermodynamic, wind,
and surface properties under a relatively pristine aerosol scenario, by running the same large model ensemble under more
polluted conditions. We use the term "polluted" here simply to mean increased aerosol loading and are not necessarily referring
specifically to anthropogenic aerosols. Comparing these two large model ensembles allows us to assess the role of radiatively
and microphysically active aerosols on sea breeze convection and how these aerosol effects may change as a function of the

wide range of tropical sea breeze environments represented here.

    Detailed experiment design, model configuration, and analysis methodologies are demonstrated in Section 2. Section
3 describes the development of sea breeze convective systems, including convection developing ahead of and along the sea
breeze front. The overall differences between pristine and polluted ensembles in terms of radiation and associated convective
environments are then shown in Section 4. In Section 5, changes in convective responses represented by cloud top heights,

updraft velocities, and surface precipitation with enhanced aerosol loading are discussed. The environmental modulation of
the various aerosol impacts on the sea breeze convective system is simultaneously examined throughout Sections 4 and 5 and
summarised in section 6.

## 2 Experiment design and analysis methodologies

### 2.1 RAMS model configuration

Two large ensembles of idealised numerical simulations of tropical sea breezes are conducted using the Regional
Atmospheric Modelling (RAMS) version 6.2.08 (Cotton et al., 2003; Saleeby and van den Heever, 2013). Idealised simulations
are useful in that they are sufficiently complex to capture the storm systems of interest but are also sufficiently simple to allow
for the isolation and evaluation of the critical physical processes at play, without the addition of unnecessary confounding
factors often present in more complex case-study simulations. As our focus is on tropical sea breeze convection, the idealised

simulations are initialised using conditions that are representative of equatorial coastal rainforest regions, more specifically,
the Cameroon rainforest region (Grant and van den Heever, 2014).

    The RAMS model configuration used is here is identical to that used in Park et al. (2020). The boundary conditions,
initialisation, and physical schemes are summarised in Table 1. The horizontal grid spacing is 1 km, and a 100 m vertical grid
spacing near the surface is vertically stretched to 1 km near the model top. With this spatial resolution, the structure of the sea

breeze circulation and associated convective clouds are well represented (Park et al., 2020). Each simulation is run from 0000
Local Time (LT) for 24 h using a 3 s time step. Output files are saved every ten minutes. RAMS is coupled to the Land-





Ecosystem–Atmosphere Feedback version 3 (LEAF–3), a fully interactive soil–vegetation–atmosphere parameterization (Lee, 1992; Walko et al., 2000). To simulate an idealised sea breeze circulation, two different surfaces, one land and one ocean, are separated by a straight coastline located at the centre of the domain. In our idealised setup, the western half of the domain

represents the land region and is specified to be a rainforest with evergreen broadleaf trees and sandy clay loam soil type, following that of Grant and van den Heever (2014). The eastern half of the domain is over ocean, and the sea surface temperature and horizontal gradient are kept fixed throughout the simulation. These relative locations of land and ocean are chosen arbitrarily and could have just as easily been the other way around.

All of the simulations are initialised with horizontally homogeneous thermodynamic and wind profiles. As described

in Park et al. (2020), we make use of 130 different initial environmental conditions, where ten different lower-tropospheric thermodynamic, wind, and surface properties (Table 2) are simultaneously perturbed across a range of values representative of tropical equatorial regions, and which were sourced from the sea breeze literature (Igel et al., 2018). These ten parameters are perturbed using maximin Latin Hypercube sampling (Morris and Mitchell, 1995), a space-filling algorithm, to ensure optimal coverage of the ten-dimensional parameter space with a minimum number of parameter combinations, and hence

determines the minimum number of simulations that need to be conducted. As shown in Park et al (2020), this amounts to 130 simulations in each of the ensembles conducted here. It should be emphasised that these 130 different initial conditions represent more than 59,049 ($=3^{10}$) different initial conditions had only three values for each of the ten parameters been selected. For the lower-tropospheric thermodynamic properties, five parameters defining the structure of the boundary layer and inversion layer are considered: (1) the boundary layer potential temperature; (2) the boundary layer relative humidity; (3) the

boundary layer height; (4) the inversion layer depth; and (5) the inversion layer strength. The upper tropospheric thermodynamic profiles above the inversion layer are identical for all 130 initial conditions and utilize the initial sounding of Grant and van den Heever (2014). The initial zonal wind speed without vertical shear is also considered and represents the sixth environmental factor examined. Finally, sensitivity to the variations in the surface characteristics are also examined including: (7) the soil saturation fraction; (8, 9) the temperature difference between land/ocean surface and the atmospheric

temperature at the lowest model level; and (10) the horizontal gradient of sea surface temperature (SST). The ranges of the ten parameters tested are based on previous studies, as described in Igel et al. (2018) and Park et al. (2020), and are also shown in Table 2.

For the sake of simplicity, the aerosol type utilised in this study is restricted to single-mode, submicron, ammonium sulphate. We chose ammonium sulphate due to its ubiquity in the atmosphere and its ability to serve as CCN as a result of its

high solubility. While the aerosol field is initialised horizontally homogeneously using appropriate profiles of aerosol (as discussed below), aerosol particles are allowed to be redistributed via advection, convection, and nucleation following initialization. Aerosol sources and sinks, including the return of aerosols to the environment following evaporation, are all also incorporated (Saleeby and van den Heever, 2013).





## 2.2 Experiment setup

Two different ensembles of idealised simulations with different aerosol concentrations are performed for this analysis. The nomenclature and description of each of the ensembles are summarised in Table 3. Here, "r" refers to aerosol–radiation interactions, and "On" means aerosol particles are allowed to interact with radiation. It should be noted that radiation is always fully interactive with both aerosols and hydrometeor species in both of the ensembles analysed here. We have conducted additional ensembles in which we have turned the aerosol-radiation interactions off to investigate the impacts on the deep

convective mode, the results of which will be published in a subsequent paper. The rOn-500 ensemble is the suite of simulations examined in Park et al. (2020) in which the model is initialised with surface aerosol concentrations of 500 mg$^{-1}$ (blue line) and which decrease exponentially with height, as shown in Figure 1. To examine the impacts of increasing the aerosol loading on sea breeze convection, this ensemble of simulations is repeated, but with aerosol concentrations of 2000 mg$^{-1}$ (red line in Figure 1) at the surface, and again using an exponentially decreasing profile. This model ensemble is referred to as "rOn-

2000". The surface aerosol concentrations are chosen based on observations made in equatorial Africa (Andreae et al., 1992; Kacarab et al., 2020). The exponentially decreasing initial profiles are used in the interests of simplicity, and they soon evolve as a result of the processes described above. We shall use the terms "pristine" and "rOn-500" interchangeably, and "polluted" and "rOn-2000" interchangeably, throughout this manuscript. It is important to note that while 130 different initial conditions are implemented in each ensemble, that the polluted and pristine ensembles utilise the same 130 different initial conditions,

thereby facilitating direct comparisons between the corresponding pristine and polluted ensemble pairs. Also, as the 130 initial environmental conditions represent the ten-dimensional input parameter space, these pristine and polluted ensembles allow us to evaluate the ways in which direct and indirect aerosol impacts on sea breeze convection are modulated by the large number of different environments that support the development of such sea breeze regimes.

## 2.3 Analysis methodology

In a manner similar to Park et al. (2020), we apply an advanced statistical algorithm developed by Lee et al. (2011) and Johnson et al. (2015) that includes statistical emulation (O'Hagan, 2006) and variance-based sensitivity analysis (Saltelli et al., 1999), over the ten-dimensional input parameter space. Due to its computational efficiency, this advanced statistical algorithm has been successfully utilised in several modelling studies that quantify the sensitivity of numerical model responses to a range of input parameters (Feingold et al., 2016; Igel et al., 2018; Wellmann et al., 2018, 2020; Glassmeier et al., 2019;

Marshall et al., 2019; Park et al., 2020). This algorithm incorporates the Gaussian process emulation (O'Hagan, 2006; Rasmussen and Williams, 2006) to build a statistical surrogate representation of complex cloud-resolving model responses of the parameters of interest. Over the ten-dimensional parameter space, the emulator estimates the cloud-resolving model responses of interest at untried input parameter combinations, thereby densely sampling the output of interest and thus allowing us to understand the relationship between the perturbed input parameter and output responses, without having to conduct the

simulations representative of each set of perturbed parameters. We can then quantify the relative importance of the perturbed



input parameters on the output of interest via variance-based sensitivity analyses. Further details of how this approach is used to determine the predominant environmental factors impacting tropical sea breeze convective systems are included in Park et al. (2020). We now begin our analysis by examining the morphology of the convection that develops within the pristine and polluted ensemble of simulations.

## 3 Basic description of the sea breeze simulations


In this section, an overview of the convective morphology and development within our tropical sea breeze simulations is provided. A sea breeze circulation develops after sunrise (0600 LT), and convergence along the leading edge of the sea breeze front becomes evident at the coastline, where the highest land-sea thermal contrast is established (not shown). Throughout the daytime hours, and even shortly after sunset (1800 LT), the sea breeze front continues to propagate further

inland, which is in keeping with classical theory of such baroclinic circulations. The sea breeze front is observed to develop in all 260 simulations comprising the rOn-500 and rOn-2000 ensembles (130 simulations in each ensemble), the location of which is detected at every output timestep using an identification algorithm developed by Igel et al. (2018).

Two types of convection are evident over land between 1200–1800 LT (Figure 2), as is often the case with tropical sea breeze systems. Ahead of the leading edge of the sea breeze, daytime heating and mixing induce cumulus convection,

whereas along the leading edge of the sea breeze front, where low-level air parcels may be lifted to the level of free convection through low-level convergence, sea breeze-initiated deep convection may occur. As shown in Park et al. (2020), the strength of the sea breeze circulation and the associated convergence along the leading edge vary strongly as a function of the initial environmental conditions. Such variations result in a range of shallow through deep convective clouds being produced in association with the sea breeze front. The left column of Figure 2 displays one example from the rOn-2000 ensemble, where

both the daytime cumulus convection ahead of the sea breeze and the convection developing along the sea breeze front are shallow (cloud top heights < 4 km AGL), and remain so throughout the simulation. While both types of convection remain shallow in this scenario, the sea breeze-initiated convection is characterised by slightly deeper clouds and stronger updrafts than the daytime cumulus convection forming ahead of this line. In the right column of Figure 2, the development of deep convection (cloud top heights > 7 km AGL) along the sea breeze front in another ensemble member is shown. In this case, the

convection initiated by the sea breeze is significantly deeper than the previous case, and this sea breeze-initiated "deep mode" is accompanied by significantly heavier precipitation and stronger updrafts than those associated with the sea breeze-initiated shallow modes. The vast majority of the ensemble members fall into "shallow mode" scenario, whereas only a handful of the ensemble members display "deep mode" convection. Throughout the rest of the manuscript we will refer to the convection developing out ahead of the sea breeze front as "*daytime cumulus convection*", and to the convection developing along the sea

breeze front as the "*sea breeze-initiated convection*". The terms "shallow mode" and "deep mode" will be used to distinguish the different types of sea breeze-initiated convection. We now examine the impacts of aerosols on the developing convective environments (Section 4) and the resulting convection both ahead of, and along, the sea breeze front (Section 5).



## 4 Aerosol impacts on the convective environment

In order to examine the impacts of aerosols on the convective environment, we first examine the effects of varying
aerosol concentrations on the surface radiation budget. As ammonium sulphate aerosols are allowed to scatter and absorb
radiation as a function of the wavelength, median radius, and relative humidity within RAMS (Saleeby and van den Heever,
2013), less incoming solar radiation is expected to reach the surface. These aerosol direct effects are addressed by analysing
the differences in surface downwelling shortwave and surface upwelling longwave radiation between rOn-500 and rOn-2000
for clear-sky columns. To identify the clear-sky columns, we have assumed that the total condensate mixing ratio is smaller
than 0.01 g kg$^{-1}$ at all vertical levels within the column. As the aerosol number concentrations are increased between the pristine
and polluted ensembles, the clear-sky downwelling shortwave radiation decreases throughout the atmosphere over land and
ocean, with the maximum difference occurring at the surface for all 130 corresponding pairs of simulations of the rOn-500
and rOn-2000 ensembles (Figures 3a, b). The daytime-averaged (0600–1800 LT) surface shortwave radiation difference
between rOn-500 and rOn-2000 is 81.2 W m$^{-2}$ for the ensemble average, and has a standard deviation of 6.5 W m$^{-2}$.

The surface upwelling longwave radiation (Figures 3c and 3d) reflects the aerosol-induced changes in incoming solar
radiation. Due to the much lower heat capacity of the land surface compared to the ocean surface, the longwave emission
significantly decreases over the interactive land surface as it rapidly responds to the reduction in shortwave radiation in the
polluted ensemble. With less surface upwelling longwave radiation, the air above the land becomes cooler (Figure 3e) for the
conditions of enhanced aerosol loading compared to the cleaner conditions. Over the ocean regions, the SST is fixed throughout
the duration of each ensemble member simulation (although it can be different between ensemble members based on the
environmental parameter being perturbed). As such, the potential temperature of the lowest level air does not change
significantly over the ocean since the longwave radiation emitted from the surface remains almost the same (Figure 3f).
However, even if the SST had been allowed to be interactive, given the higher heat capacity of the ocean, the upwelling
longwave radiation from the ocean surface would not respond as quickly to the aerosol-induced reduction in shortwave
radiation as that over the land surface. As a result of these aerosol interactions with the radiation, the ensemble-mean surface
temperature difference between the ocean and the land is less in the polluted case when compared with the pristine case
(Figures 3e and 3f), which has important implications for the strength of the thermally driven sea breeze circulation. These are
discussed in more detail below.

The aerosol-induced reduction in incoming solar radiation reaching the surface also impacts the surface fluxes. Figure
4 displays the temporal evolution of the surface sensible and latent heat fluxes of the environment ahead of the sea breeze
front, averaged over all 130 simulations in each ensemble. Both the ensemble-mean sensible and latent heat fluxes are reduced
in rOn-2000 compared with rOn-500 as a result of the enhanced aerosol loading. The maximum aerosol-induced reduction is
45% and 60%, and the mean is 19% and 15%, for the sensible and latent heat fluxes, respectively. This reduction in incoming
solar radiation and associated surface flux responses due to enhanced aerosol loading is in close agreement with the findings
from previous studies (Yu et al., 2002; Koren et al., 2004; Feingold et al., 2005; Jiang and Feingold, 2006; Zhang et al., 2008;



Grant and van den Heever, 2014). These reductions in sensible and latent heating will negatively impact the convective boundary layer by limiting the heating and moistening of this layer. The surface-based mixed layer depth, defined here as the level above the surface at which the vertical gradient of the potential temperature first exceeds 2 K km$^{-1}$, decreases in rOn-2000 due to this reduction in surface sensible heat flux and associated turbulent mixing. The percentage difference of the mean

surface based-mixed layer depth ahead of the sea breeze front between rOn-500 and rOn-2000 is shown in Figure 5a, where it is evident that the mixed layer in each member of the polluted ensemble is shallower than the corresponding mixed layer in the pristine ensemble, with differences ranging from 1.6% to 23%.

         The results of this section thus far indicate that two of the three convective ingredients (Doswell, 1987), lower-tropospheric moisture (as represented by the latent heat fluxes) and instability (as represented by the mixed layer depth),
become less favourable for convective development in all of the rOn-2000 members compared with rOn-500 members, as a result of the aerosol-induced reduction of incoming solar radiation. The third convective ingredient is vertical lift. In these simulations, lift is provided both by the sea breeze circulation itself and the convergence along the sea breeze front. Classical sea breeze theory dictates that, to first order, the faster the sea breeze moves, the further inland the sea breeze travels during the day and the stronger the convergence along the sea breeze front, and hence the greater the vertical lift along the front. Here
we examine the maximum inland extent of the sea breeze front as an indicator of sea breeze intensity and hence vertical lift (Figure 5b). The maximum inland extent of the sea breeze front is identified as the last inland location of the sea breeze front detected by the sea breeze front algorithm (Igel et al., 2018). The percentage difference of the maximum sea breeze inland extent between the polluted and pristine ensembles varies from −2% to −53% across the ensembles, and it is evident from Figure 5b that the sea breeze extent is significantly less in rOn-2000 than rOn-500 for each and every one of the ensemble
pairs. It is therefore evident from these results that the third primary element of convection, vertical lift, is also reduced with increasing aerosol loading. As all three convective ingredients (moisture, instability, and lift) are reduced in association with aerosol direct effects on the sea breeze circulation and the convective boundary layer, we could therefore expect the presence of increased aerosol concentrations to reduce the intensity of the sea breeze convective system in all of the different environments represented by the ensemble members. However, this assumes that aerosol indirect effects do not significantly
contribute to convective storm intensity, thereby offsetting some of these direct effects. We investigate these effects in more detail in the next section.

         Finally, a reduction in the three convective ingredients with enhanced aerosol loading was also observed in Grant and van den Heever (2014). However, in that study, they only utilised one set of initial environmental conditions. Therefore, one might wonder whether such findings are applicable to the multiple other environments known to support sea breeze systems,
or whether various sea breeze environments produce a different response to aerosol loading. Since rOn-500 and rOn-2000 each have 130 different initial conditions, and as these initial conditions are representative of the wide range of environmental parameter values previously identified as being responsible for generating sea breeze systems, these results suggest that the aerosol-induced weakening of sea breeze-initiated convection through direct effects is indeed a robust result, and one that occurs irrespective of the sea breeze environment.





## 5 Impacts of enhanced aerosol loading on continental convection


In the previous section, we demonstrated that the three convective ingredients of moisture, instability, and lift, are reduced in the polluted ensemble, when compared with the pristine ensemble, and that this is due to the direct impacts of enhanced aerosol loading on incoming solar radiation. Now we seek evidence of the impacts of increased aerosol number concentrations on the intensity of the convection within tropical sea breeze convective system as a result of both direct and

indirect aerosol effects. More specifically, we investigate the impacts of enhanced aerosol loading on cloud top heights, convective updraft velocities, and precipitation, all of which are common indicators of convective intensity (e.g., Nesbitt and Zipser, 2003).

### 5.1 Cloud top heights

First, we examine the impacts of aerosol loading on cloud top height by analysing the following: (1) the frequency

distribution of the low (cloud top height < 4 km AGL), middle (cloud top heights between 4–7 km AGL), and deep (cloud top heights > 7 km AGL) clouds and (2) the maximum cloud top height during the afternoon hours (1200–1800 LT) anywhere over land. This includes clouds both ahead of and along the sea breeze front. To identify cloud top heights, we start at the bottom of each model column and locate the lowest and highest level where the total condensate exceeds 0.1 g kg$^{-1}$. This column is then marked as a cloudy column when all the points between these two levels contain total condensate mixing ratios

greater than 0.1 g kg$^{-1}$. We regard the highest level of the cloudy column as the convective cloud top height. This check on cloud contiguity means that we do not inadvertently take into account those situations in which there may be multiple clouds at different levels within the column.

As shown in Figure 6a, the vast majority of ensemble members in the pristine ensemble are dominated by low clouds, with cloud tops less than 4 km AGL. In other words, only shallow convective clouds develop both ahead of and along the sea

breeze front in most of the ensemble members, and while all of the environmental conditions tested here support the development of sea breeze convection, most do not support the development of the sea breeze-initiated deep convective mode. This low cloud dominance is also observed for the polluted ensemble (Figure 6a). Specifically, among the 130 simulations in each ensemble, there are 104 and 113 simulations with low clouds only in rOn-500 and rOn-2000, respectively.

The suppression of sea breeze convective intensity in rOn-2000, when compared with rOn-500, is evident in the

maximum cloud top heights (Figure 6c), as well as in the percentage difference in the maximum cloud top height (Figure 6d). It is clear from these figures that the maximum cloud top heights are always less in rOn-2000 than in rOn-500, except for three sea breeze-initiated deep mode cases (Tests 27, 79, and 104). In other words, the maximum cloud top height of the sea breeze-initiated shallow mode and the daytime cumulus convection always decreases with more aerosol loading, regardless of different initial conditions. For the sea breeze-initiated deep mode cases, however, the sign of the maximum cloud top height

response to enhanced aerosol loading varies with the initial environmental conditions. Furthermore, Figure 6c demonstrates that while there are 12 cases (Tests 27, 29, 37, 41, 59, 65, 75, 79, 100, 104, 106, and 120) with a deep mode in rOn-500, only





7 of these 12 cases (Tests 27, 59, 65, 75, 79, 104, and 120) have a deep mode when the aerosol loading is enhanced in rOn-2000. The reasons for this are discussed below.

## 5.2 Convective updraft velocities

In order to better understand the dynamical response of convection to enhanced aerosol loading, updrafts developing over land between 1200 and 1800 LT with velocities greater than 1 m s$^{-1}$ are analysed. The maximum updraft velocity (Figure 7a) represents the intensity of the most vigorous continental convection. In all of the simulations of the pristine and polluted ensembles, the strongest updrafts are always found in association with the convergence along the sea breeze front, irrespective of whether the convection produced by this process is shallow or deep. Aerosol-induced suppression of the maximum updraft

velocities is evident in most of the member pairs (Figure 7a), however, there are 10 ensemble pairs (Tests 18, 28, 36, 67, 76, 79, 83, 84, 111, and 124) in which the maximum updraft velocities are stronger in rOn-2000. As such, enhanced aerosol loading may therefore produce weaker or stronger maximum updraft velocities in the sea breeze-initiated convection. As this aerosol-induced response varies as a function of the initial environmental conditions, it is environmentally modulated. The mean (Figure 7b) and lower percentiles (25$^{th}$, 50$^{th}$, 75$^{th}$, 95$^{th}$) (Figures 7c-f) of the updraft velocities, on the other hand, also include

the updraft velocities of the daytime cumulus convection that forms ahead of the sea breeze front, and in fact are dominated by the convection forming in this region (Park et al., 2020). All 130 ensemble pairs demonstrate a reduction in both the mean and 25$^{th}$ through 95$^{th}$ percentile updraft velocities with enhanced aerosol loading. This ubiquitous signal of weakening daytime cumulus updrafts is perhaps not surprising considering the aerosol-induced reduction in the convective mixed layer depth and surface fluxes presented in Section 4. Furthermore, as the mean and lower percentiles are dominated by the daytime cumulus

convection developing ahead of the sea breeze, this suggests that aerosol-induced weakening of the updraft velocities of the convection ahead of the sea breeze front occurs irrespective of the environmental conditions, and is therefore a robust signal.

        In the following subsections, we now seek to determine whether the key environmental parameters identified in Park et al. (2020) as the predominant parameters driving the convective updrafts in rOn-500 are impacted by aerosol loading, and if so, why that is the case. We first examine the sea breeze-initiated convection, followed by the daytime cumulus convection

developing ahead of the sea breeze front. We examine these two types of convection separately as they are driven by different processes, with convergence and convective instability along the sea breeze front being critical to the sea breeze-initiated convection, and daytime surface heating, boundary layer mixing, and thermal buoyancy driving the convection developing ahead of the sea breeze.

### 5.2.1 Sea breeze-initiated convective updrafts

Due to the steep vertical velocity gradients between sea breeze-initiated shallow and deep convective modes in rOn-500, Park et al. (2020) found that constructing a statistically robust emulator for the maximum updraft velocities was not feasible. The same is true here for the rOn-2000 simulation, and thus we have had to rely on other analysis methods to determine why deep convection is absent in 5 out of 12 cases in the polluted scenario. As noted in Park et al. (2020), the 12 cases with



sea breeze-initiated deep modes in rOn-5000 all have initial boundary layer potential temperatures greater than 297 K as a
common parameter. Given the range of initial boundary layer potential temperatures tested in this study (Table 2), this critical
threshold for the deep mode falls at the upper end of this range and is the reason for why the vast majority of ensemble members
have shallow modes. Figure 8 is a scatter plot relating various convective characteristics to the initial boundary layer potential
temperature, for all 130 ensemble members. It is clear from this figure that the sea breeze-initiated deep convective mode in
rOn-500 occur in all of the ensemble members in which the initial boundary layer potential temperature is 297 K or greater
(Figures 8c and 8e), and in which the mixed layer CAPE is greatest (Figure 8a). A similar dependence on the initial boundary
layer potential temperature threshold is shown for rOn-2000 (Figures 8d and 8f). However, in rOn-2000 the threshold above
which the deep convective mode occurs is 299 K, 2 K greater than that in rOn-500. As demonstrated above, the surface
temperatures are reduced in the presence of enhanced aerosol loading in the rOn-2000 ensemble, thereby reducing the
convective potential in these simulations. Therefore, greater initial boundary layer potential temperatures are necessary to
offset the aerosol-induced surface cooling in the polluted ensemble, thereby providing sufficient environmental CAPE (Fig.
8b) to support the production of deep convection along the sea breeze front.

### 5.2.2 Daytime cumulus convection updrafts

Unlike the sea breeze-initiated convective updrafts, constructing a robust emulator was possible for the updraft
velocities of the daytime cumulus convection forming ahead of sea breeze front. We can therefore draw on the emulator results
in our analysis of these updrafts. The bar graphs in Figures 9a (rOn-500) and 9b (rOn-2000) indicate how much of the variance
in the median updraft velocity is explained by the individual perturbations to the ten environmental parameters tested. Each
stacked bar graph's height refers to the summation of first-order contributions of the ten parameters to the median updraft
velocity. Any blank space remaining above the bar indicates the contributions made by higher-order nonlinear interactions
involving multiple parameters. It is evident from the left most bar graph (labelled "Overall") that the same two parameters, the
soil saturation fraction (dark grey) and the inversion layer strength (pink), are the predominant contributors to the median
updraft velocities in rOn-500 and rOn-2000, although their percentage contributions differ. In order to understand the impacts
of aerosols on these relative contributions, we first need to understand the processes driving these two predominant
contributions.

Figures 9c and 9d present the mean responses of the emulator-predicted median updraft velocities to the soil saturation
fraction and inversion layer strength. It is clear from these figures that drier soils and weaker inversion layers promote more
vigorous daytime cumulus convection in both the pristine and polluted regimes. While the response of the updraft velocities
to the strength of the inversion layer is a relatively simple, continuously decreasing function, the sensitivity of the median
updraft velocities to the soil saturation fraction (Figure 9c) shows three relatively distinct regimes: (1) moderate velocity
changes for soil saturation fractions between 0.1–0.4; (2) large velocity changes for soil saturation fractions between 0.4–0.6;
and (3) relatively neutral velocity responses for soil saturation fractions between 0.6–0.9. We will refer to these three regimes
as the DRY, MID, and WET soil moisture regimes based on the response curve in Figure 9c. The corresponding variance-





based sensitivity analyses for the daytime cumulus convection stratified by these three different soil regimes are presented in Figures 9a and 9b (second to fourth bar graphs from the left).

To understand the median updraft velocity responses in the soil moisture regimes, one needs to note that the soil
saturation fraction value of 0.4, which separates the DRY and MID soil moisture regimes, corresponds to the permanent wilting point of sandy clay loam soil. The permanent wilting point is the minimum amount of soil moisture that a plant's roots require in order not to permanently wilt. Below the permanent wilting point, the vegetation becomes stressed, resulting in a shutdown of evapotranspiration and thereby suppressing the release of water vapor (Hohenegger and Stevens, 2018; Drager et al 2020). In the DRY soil regime (0.1–0.4), where the soil moisture falls below that of the permanent wilting point, the surface latent
heat flux is suppressed (Figure 9f), and more of the surface heating goes into enhancing the sensible heat fluxes (Figure 9e). The relatively strong sensible heat fluxes in the DRY regime contribute to warmer, deeper boundary layers and hence the stronger updraft velocities observed in this regime (Figure 9c). Over the WET soil regime (0.6–0.9) with abundant evaporation and transpiration, the surface sensible heat flux is reduced and the surface latent heat flux is enhanced (Figures 9e and 9f) when compared with the DRY soil regime. Furthermore, the surface sensible heat flux is no longer sensitive to the soil saturation
fraction, as is evident in the flat gradient of the curve in Figure 9e, as well as in the absence of the soil saturation fraction contributions to the WET regime in Figures 9a and 9b (fourth bar graph from the left). The updrafts within the WET soil regime are therefore weaker. Finally, in the MID soil regime, both the sensible and latent heat fluxes show the most sensitive responses to changes in the soil saturation fraction (Figures 9e and 9f). While the median updraft velocities (Figure 9c) and surface sensible heat flux (Figure 9e) are similar in trend to those over the DRY soil regime, the slopes are steeper in the MID
soil regime. As such, the relative importance of the soil saturation fraction in contributing to updraft velocity variance is greatest in the MID soil regimes compared with the DRY or WET soil regimes (second to fourth bar graphs from the left in Figures 9a and 9b). Drager et al. (2020) also noticed non-linear responses to soil moisture focused around the permanent wilting point in their study of soil moisture impacts on cold pools.

We now turn to the impacts of enhanced aerosol loading on the roles of the soil saturation fraction and the inversion
layer strength. When comparing Figure 9a with 9b, the relative percentage contribution of soil saturation fraction to updraft variability decreases from 78% to 68% with enhanced aerosol loading, whereas that of the inversion layer strength increases from 4% to 8%. The soil saturation fraction plays an important role in the daytime cumulus convection ahead of the sea breeze front through its control of the magnitude of the surface sensible and latent heat fluxes, and its partitioning between them. As discussed in Section 4, the aerosol-induced reduction in surface downwelling shortwave radiation in the polluted ensemble
reduces the incoming shortwave radiation, the surface longwave emission, the associated surface sensible and latent heat fluxes, and the mixed layer depth. The reduction in the percentage contributions of the overall soil saturation fraction to the median updraft velocities in the polluted ensemble (compare left bars in Figures 9a and 9b) therefore reflects this reduced role of the surface fluxes and boundary layer mixing in driving the updrafts. When examining aerosol impacts on the specific soil moisture regimes (three bars on the right of Figures 9a and 9b) it is clear that under conditions of enhanced aerosol loading the
relative importance of soil saturation fraction on the median updraft is reduced in the DRY and MID soil regimes, while it is





completely absent in WET soil regimes regardless of aerosol loading. Comparing the emulator-predicted median updrafts between the pristine and polluted conditions (Figure 9c), the trends in the relationship between the soil saturation fraction and the updraft velocities are very similar, but median updrafts are stronger in rOn-500 due to the stronger sensible heat fluxes.

In rOn-500, inversion layer strength is the second-most important parameter for the mixed-layer depth and the median updraft velocity. When the initial inversion layer is weaker, the lower troposphere is less stable, which promotes daytime turbulent mixing and leads to stronger vertical motions. While the inversion layer is still the second-most important parameter for the median updraft in rOn-2000, its relative importance is increased from 4% to 8%. As the contributions made by the rest of the factors to the velocity variance remains much the same between the pristine and polluted conditions (Figures 9a and b), this increase primarily reflects the reduced contribution by the soil saturation fraction and the concomitant increase in the

contribution of the inversion layer strength to boundary layer development, and hence, to boundary layer updrafts.

       To summarize, the daytime cumulus updraft velocities become weaker with enhanced aerosol loading. This is consistent with the reduction in surface fluxes and mixed layer depth under polluted conditions described in section 4. The ten-dimensional variance-based sensitivity analysis described here demonstrates that the relative importance of the soil saturation fraction and inversion layer strength in contributing to the updraft velocity variance varies as a function of aerosol

loading. The degree of aerosol-induced weakening of the updrafts is thus modulated by the aerosol-environmental feedbacks.

### 5.3 Surface accumulated precipitation

       The changes in surface accumulated precipitation with increased aerosol loading are now analysed. Figures 10a and 10b display land-averaged accumulated surface precipitation at 1800 LT in rOn-500 and rOn-2000, respectively. The impact of enhanced aerosol loading on precipitation is represented by the percentage differences between rOn-2000 and rOn-500

(Figure 10c). The percentage differences are only determined for those simulations that produce at least 0.1 mm of land-averaged accumulated precipitation in rOn-500 (36 out of 130 simulations). Only 19 simulations produce more than 0.1 mm of area-averaged precipitation in both rOn-500 and rOn-2000, and no simulation in rOn-2000 produces more than 0.1 mm of precipitation when its corresponding ensemble member in rOn-500 does not produce precipitation. Figure 10c distinctly shows that the accumulated precipitation is reduced in all 36 of the rOn-2000 precipitating ensemble members when compared with

their corresponding counterparts in rOn-500, the percentage differences of which vary from −16.3% (Test75) to −96.7% (Test110).

       We now examine the differences in the bulk microphysical processes contributing to the differences in the surface precipitation in the 36 precipitating ensemble pairs (Figure 11). The following five processes contribute to the production of precipitation at the surface:

1) cloud-to-rain: cloud water transferred to rain through collection;

       2) vapor-to-rain or rain-to-vapor: vapor diffusion onto rain (gain term) or evaporation from rain (loss term);

       3) melting of ice: ice mass transferred to rain via thermodynamic melting as the ice species fall below the freezing level;



4) ice-to-rain: collisional ice melting due to collection of warmer rain; and

5) rain-to-ice: rainwater collected by ice species through riming.

These process rates are averaged over the land domain between 1200 and 1800 LT. Since the cloud frequency over land is heavily weighted by the daytime cumulus convection mode, the averaged mixed-phase process contributions are relatively small. As most of the differences in the processes contributing to differences in surface rainfall occur below 10 km AGL, we restrict the axes in Figure 11 to between the surface and 10 km AGL to enhance figure clarity. It should also be noted that the

freezing level varies from simulation to simulation due to the different initial temperature and moisture profiles. It is evident that in all simulations, regardless of the different initial conditions, the average warm rain production rates (i.e., cloud-to-rain) are greater in rOn-500 than rOn-2000. Similarly, average rain evaporation rates (i.e., rain-to-vapor) are also greater in magnitude in rOn-500 than rOn-2000, although they are less than the cloud-to-rain production term. In all of the ensemble pairs, except for Test68 and Test120, where the ice-to-rain process is the most dominant gain term, the warm rain production

rates are greater than cold rain production rates throughout the whole profile. Figure 11 therefore demonstrates that, in the mean, precipitation suppression in the polluted environment primarily occurs as a result of the suppression of warm rain formation. Furthermore, as this response is evident across 34 out of 36 precipitating members, it appears that these warm rain trends are independent of the environmental conditions, although the magnitudes in the response certainly do vary with environment. No common characteristics are, however, found in the changes in cold rain production with enhanced aerosol

loading. As such, our results suggest that the environment modulates cold rain response to enhanced aerosol loading.

Examining the most noticeable differences in the initial conditions of the two extrema rainfall difference cases (Test75 and Test110) provides some insights into the environmental controls on these precipitation responses to aerosol loading (Figure 12). Test75, which demonstrates the least reduction in precipitation between the pristine and polluted cases, has a very moist, warm and deep lower troposphere that develops in association with a soil saturation fraction of 0.89, a boundary layer relative

humidity of 89.8% and a boundary layer potential temperature of 299 K. On the other hand, Test110, which shows the greatest difference in precipitation between the pristine and polluted cases, has only moderately moist and warm initial conditions with a soil saturation fraction of 0.44, a boundary layer relative humidity of 84.2%, and a boundary layer potential temperature of 288 K. The warmer, moister conditions in Test75 will (a) allow for more effective aerosol nucleation and the subsequent growth of the more numerous, smaller cloud droplets that form in the polluted case compared to drier, polluted conditions, and

(b) more effectively limit the impacts of evaporation on raindrop populations than in the drier, polluted cases. As such, is appears that the warmer, moister conditions assist in muting the impacts of aerosols on the warm rain production, resulting in rain gain and loss terms that are of similar magnitude in the polluted and clean conditions, and hence relatively small aerosol-induced reductions in net rain production. Drier conditions, on the other hand, enhance the aerosol-induced differences in the rain source and sink terms through less efficient droplet growth and enhanced raindrop evaporation. Similar trends can be

found for other simulations with warm, moist boundary layers, such as Test84, and drier, colder boundary layers, such as Test17 (Figure 12). Now, when the initial boundary layer temperature and relative humidity are similar between ensemble members, then lower soil saturation appears to be associated with greater aerosol-induced precipitation suppression. For





example, Test54 and Test40 have similar boundary layer temperatures and relative humidity (Test54: 294.2 K and 92.9%; Test40: 293.1 K and 93.2%), but Test40 shows greater aerosol-induced precipitation suppression. The wetter soil in Test54

(soil saturation = 0.83) compared with that in Test40 (soil saturation fraction = 0.64) appears to limit the aerosol-induced precipitation suppression. The same could be applied to Test43 and Test59 (Test43: 298.1 K and 76.4%; Test59: 299.8 K and 76.2%) where the soil saturation fraction is higher in Test43 (soil saturation fraction = 0.68) than Test59 (soil saturation fraction = 0.57). However, the same cannot be said for all 36 precipitating pairs, and as such, the extent to which aerosols suppress precipitation therefore appears to be a complex function of different environmental variables.

In summary, under enhanced aerosol concentrations, the reduction of surface accumulated precipitation is observed irrespective of the initial environmental conditions. While the trends of aerosol-induced decreases in precipitation is robust across the sea breeze environments investigated here, the magnitude of precipitation reduction varies significantly across the ensemble members, and appears to be a relatively complex function of the environment.

## 6 Summary and discussion

The primary goals of this study have been 1) to investigate how microphysically and radiatively active aerosol particles influence the wide range of convective environments supporting sea breeze convective regimes and the convective cloud characteristics developing under these regimes, and 2) to determine whether the convective environment modulates these aerosol impacts on the convection. In order to achieve our goals, we conducted two large numerical model ensembles, where the only difference between the ensembles was the aerosol loading. Each of these two ensembles contained 130 members

initialised with 130 different initial conditions, representing the simultaneous perturbation of ten thermodynamic, wind, and surface properties. The selection of these 130 initial conditions was based on a statistical space-filling method (Morris and Mitchell, 1995) and the clean (rOn-500) and polluted (rOn-2000) ensembles were then compared using the statistical framework developed by Johnson et al. (2015).

      The comparison between rOn-500 and rOn-2000 demonstrated that aerosol direct effects resulted in less shortwave

radiation reaching the surface. The associated longwave emission from the land surface was also reduced, thereby producing a cooler land surface, a smaller land-sea thermal contrast and a weaker sea breeze circulation. The land-surface sensible and latent heat fluxes, the surface-based mixed layer depth, and the sea breeze inland extent were all subsequently reduced in the polluted scenario as a result. The three ingredients of moist convection, i.e., moisture, instability, and lift (Doswell, 1987), were all therefore reduced in the presence of aerosol direct effects. The reduction in these three convective ingredients as a

result of enhanced aerosol loading was found in all of the 130 ensemble simulation pairs, and therefore suggest a robust result that is independent of environmental conditions. The magnitude of the reduction does vary as a function of the initial environmental. These results extend those of Grant and van den Heever (2014), in which a similar sensitivity of tropical sea breeze convection to aerosol loading was demonstrated, albeit for only one set of initial conditions.



The resulting aerosol-induced changes in convection over land in the afternoon (1200–1800 LT) were then examined
for convective clouds developing along the sea breeze front. In the vast majority of ensemble members in rOn-500 and rOn-
2000 the sea breeze-initiated convection remained stronger and deeper than daytime cumulus convection forming ahead of the
sea breeze front. The shallow mode of the sea breeze-initiated convection showed an overall reduction in maximum cloud top
height and maximum updraft velocity with enhanced aerosol loading, despite different initial environmental conditions,
suggesting that the aerosol-induced suppression of shallow convective velocities is robust for the wide range of environments
tested here. However, the sign and magnitude of changes to the sea breeze-initiated deep convective velocities in response to
enhanced aerosol loading varied as a function of initial environmental conditions, with some member ensembles showing an
increase in updraft velocity, while others showed a decrease or no change to the strength of the updraft. This demonstrates that
aerosol impacts on the deep convective updrafts are environmentally modulated. Of the ten environmental parameters
perturbed in the initial conditions, the initial boundary layer potential temperature was shown to have important implications
for the deep convective mode. The sea breeze-initiated deep mode was only observed in those rOn-2000 ensemble simulations
in which the initial boundary layer potential temperatures were greater than 299 K, compared with a corresponding threshold
of 297 K in rOn-500. With enhanced aerosol loading inducing cooler and more stable near-surface environments in rOn-2000,
the warmer initial boundary layer potential temperatures were found to be necessary to facilitate deep convection through
higher CAPE.

The aerosol-induced reduction in surface fluxes and mixed-layer depth also led to the suppression of daytime cumulus
convection ahead of sea breeze front, irrespective of different initial conditions, and thus appears to be a robust response.
Furthermore, a ten-dimensional variance-based sensitivity analysis, combined with statistical emulation, revealed that the
relative importance of soil saturation fraction on the surface fluxes and mixed-layer depth, and hence on the daytime cumulus
updrafts, was reduced in the presence of enhanced aerosol loading. A nonlinear sensitivity of boundary layer updraft velocities
to soil saturation fraction was also found, with the greatest convective updraft response observed in the MID soil saturation
fraction regime, followed by a moderate response in the DRY soil saturation fraction regime and no response in the WET soil
regime. These sensitivities were found to exist in both the clean and polluted environments. The sensitivity analysis results
therefore emphasise the importance of considering atmosphere-radiation-land feedback processes in the modelling studies of
aerosol-cloud interactions.

Finally, changes in the surface-accumulated precipitation to increased aerosol loading were analysed. Again, a
consistent aerosol-induced reduction was observed across the entire ensemble of model simulations. Spatiotemporally
averaged rain production and evaporation rates exhibited robust aerosol-induced behaviours. While the trends in precipitation
reduction were found to be consistent irrespective of the environment, the magnitude of precipitation suppression was found
to be modulated by environments.

Many of the past studies assessing aerosol impacts on deep convection have tended to neglect the role of aerosol
direct forcing in order to focus on aerosol indirect effects. However, our results indicate the importance of considering both
aerosol direct and indirect effects when determining the impacts of aerosols on convective systems, and suggest that future





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



**Table 1. The RAMS model configuration used to conduct the clean and polluted ensembles**

| Model Aspect | | Setting |
|---|---|---|
| Grid | | Arakawa C grid (Mesinger and Arakawa, 1976)<br>Single grid<br>Non-rotating ($f = 0$ s$^{-1}$)<br>1500 points × 150 points, $\Delta x = \Delta y = 1$ km<br>57 vertical levels, $\Delta z = 100$ m lowest level stretched to $\Delta z = 1$ km aloft |
| Integration | | 24 hr, $\Delta t = 3$s |
| Boundary conditions | | Zonally open-radiative (Klemp and Wilhelmson, 1978), meridionally periodic |
| Initialisation | | 130 horizontally homogeneous thermodynamic and wind profiles where the ten parameters are simultaneously perturbed within the indicated value ranges (Igel et al., 2018; Park et al., 2020)<br>Random potential temperature perturbations within the lowest 500 m of the domain with a maximum perturbation of 0.1 K at the surface are used to initiate convection |
| Radiation | | Two-stream, hydrometeor-sensitive (Harrington, 1997)<br>Updated every 60s |
| Turbulence | | Smagorinsky (1963) deformation K with stability modifications (Hill, 1974) |
| Surface | Land | Two-way interactive Land–Ecosystem–Atmosphere–Feedback version 3 (LEAF-3, Walko et al., 2000)<br>Evergreen broadleaf tree with 90% vegetation fraction<br>11 vertical soil levels with sandy clay loam |
| | Ocean | Non-interactive, with fixed sea surface temperature (SST) ranging from 276.2 K to 307.6 K based on parameter perturbation (Section 2.1) and horizontal SST gradient ranging from −0.02 K km$^{-1}$ to 0.02 K km$^{-1}$ (Table 2) |
| Microphysics | | Double-moment bin-emulating bulk scheme with eight hydrometeors: cloud, drizzle, rain, pristine ice, snow, aggregates, graupel, and hail (Walko et al., 1995; Meyers et al., 1997; Saleeby and Cotton, 2004; Saleeby and van den Heever, 2013)<br>Utilise lookup tables generated offline through the use of Lagrangian parcel bin model calculations, including aerosol activation (Saleeby and Cotton, 2004), droplet collection (Feingold et al., 1988), and sedimentation (Feingold et al., 1998) |
| Aerosol treatment | | Ammonium sulphate aerosols available to act as CCN (Saleeby and van den Heever, 2013)<br>Exponentially decreasing number concentration with height from the surface; pristine = 500 mg$^{-1}$, polluted = 2000 mg$^{-1}$<br>Single mode, log-normal distribution<br>Sources and sinks<br>DeMott et al. (2010) heterogeneous ice nucleation |





**Table 2. The ten environmental parameters perturbed in this study and their uncertainty range.**

| Parameter | | Uncertainty Range |
|---|---|---|
| Boundary layer | Potential temperature | [285, 300] K |
| | Relative humidity | [75, 95] % |
| | Height | [100, 1000] m |
| Inversion layer | Strength | [1, 15] K km$^{-1}$ |
| | Depth | [100, 1000] m |
| Wind | Zonal wind speed | [−5, 5] m s$^{-1}$ |
| Sea surface | Temperature difference between the sea surface and the lowest model level atmosphere | [−10, 10] K |
| | Horizontal gradient of sea surface temperature | [−0.02, 0.02] K km$^{-1}$ |
| Land surface | Temperature difference between the land surface and the lowest model level atmosphere | [0, 10] K |
| | Soil saturation fraction | [0.1, 0.9] |




**Table 3. The naming convention and descriptions of the two model ensembles conducted for this research.**

| Name | Aerosol loading | Aerosol-radiation interactions | Total number of simulations | Colour in Figures |
|---|---|---|---|---|
| rOn-500 | pristine, 500 mg$^{-1}$ | On | 130 | Blue |
| rOn-2000 | polluted, 2000 mg$^{-1}$ | | 130 | Red |

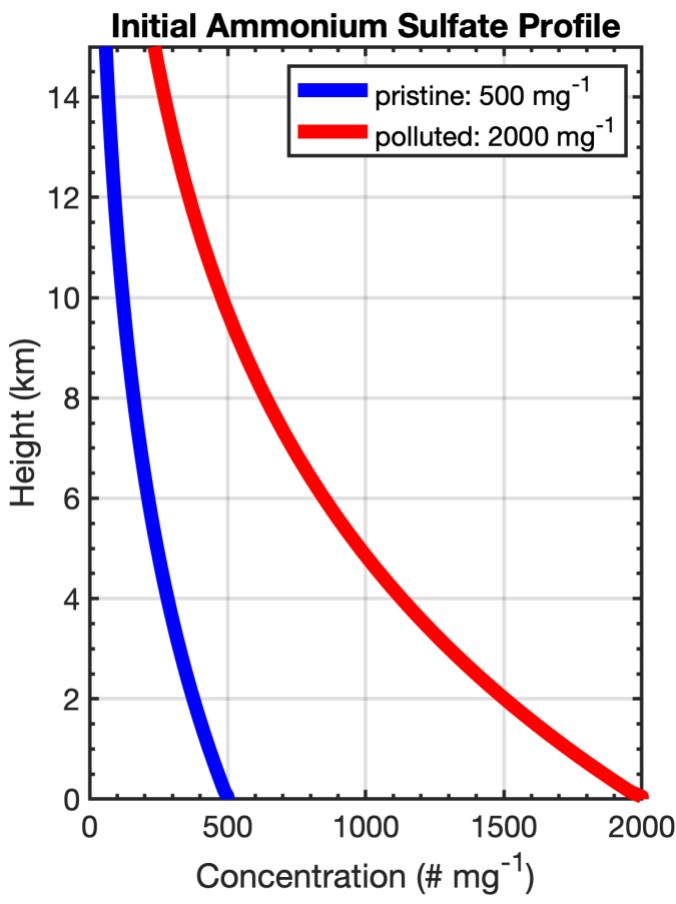

**Figure 1. Initial number concentration of ammonium sulphate for the pristine (blue line, 500 mg$^{-1}$ at the surface, rOn-500) and polluted (red line, 2000 mg$^{-1}$ at the surface, rOn-2000) model ensembles.**


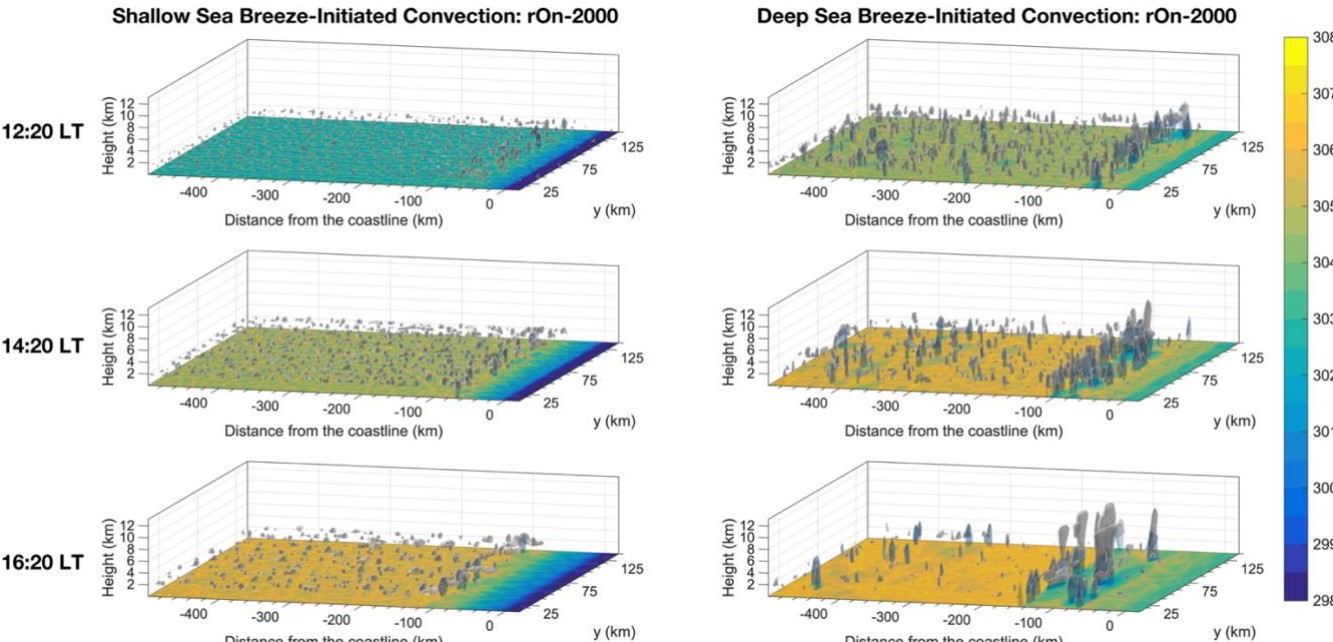


**Figure 2. Examples of the convective morphologies observed in the rOn-2000 ensemble where the sea breeze-initiated convection remains shallow throughout the domain (left column, Test 14, "shallow mode") and in which deep convection develops along the sea breeze front (right column, Test 27, "deep mode") at 1220 LT (top), 1420 LT (middle), and 1620 LT (bottom). Gray isosurfaces are where the total condensate, with the exception of rain, is 0.1 g kg⁻¹, and the dark blue isosurfaces are where rain mixing ratio is 0.1 g kg⁻¹. The shaded contours at the surface represent density potential temperature (K; Emanuel, 1994) at the lowest model level. Only a 520 km × 130 km subset of the domain is displayed here.**




**Figure 3. Time series of each ensemble-mean (a, b) surface downwelling shortwave radiation, (c, d) surface upwelling longwave radiation, and (e, f) lowest model level potential temperature, over land (left column, brown lines) and ocean (right column, blue lines). Solid and dashed lines denote rOn-500, and rOn-2000, respectively.**





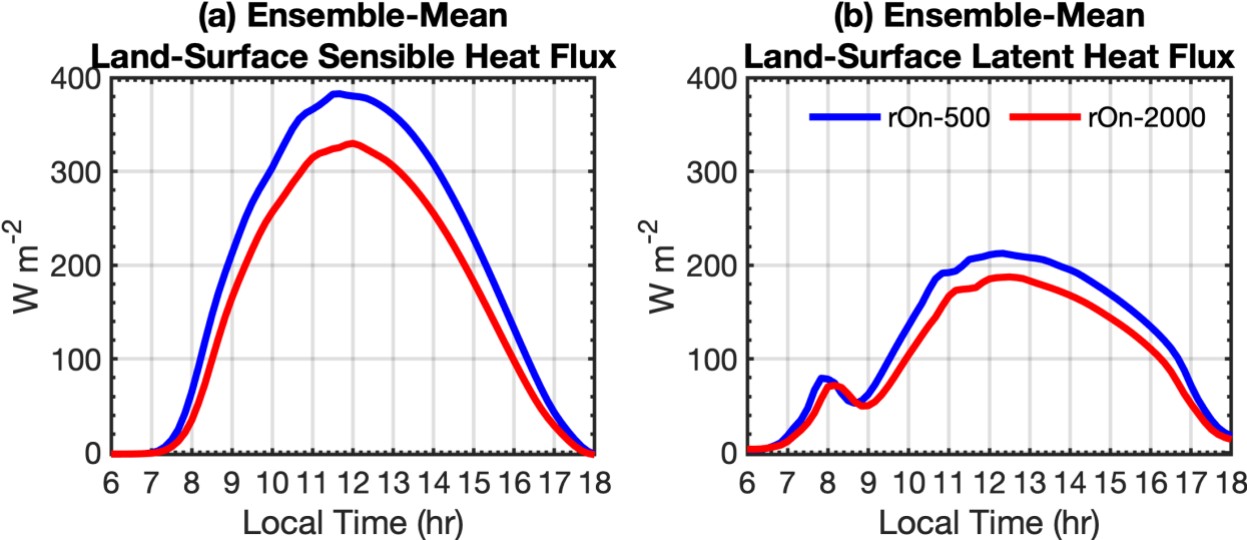

**Figure 4. Time series of the ensemble-mean land surface (a) sensible heat flux and (b) latent heat flux, spatially averaged from**
**western domain edge to the 50 km ahead of the algorithm-identified sea breeze front, for the rOn-500 (blue) and rOn-2000 ensembles**
**(red).**





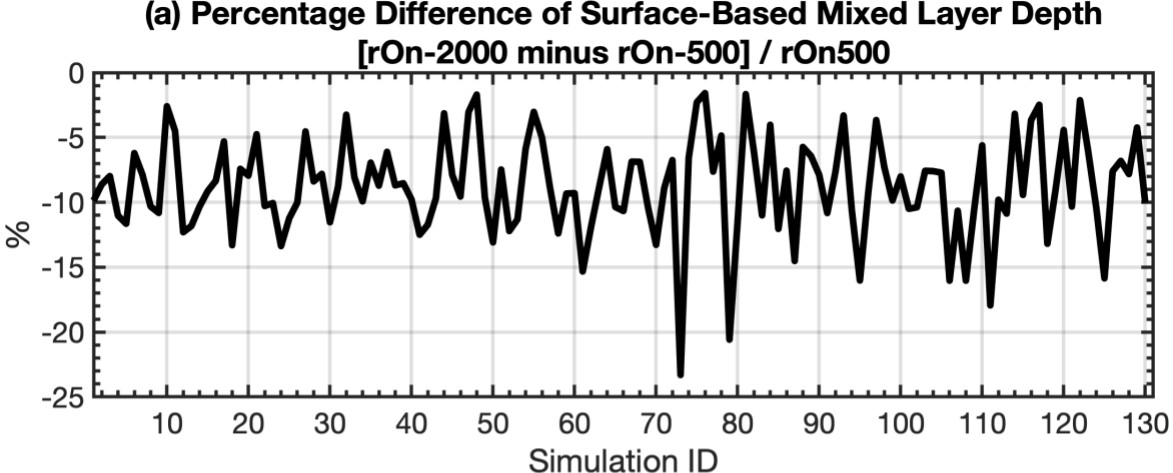

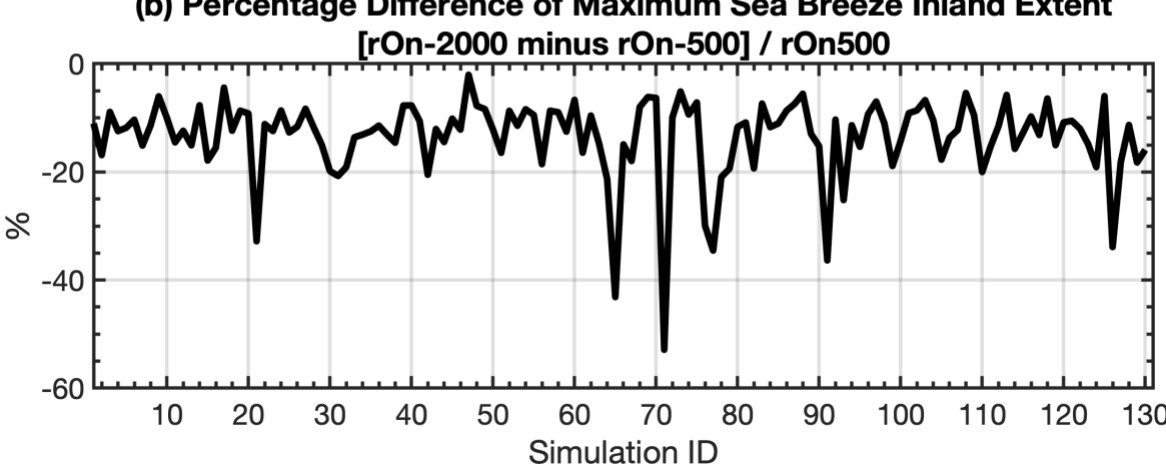

Figure 5. The percentage difference between the rOn-500 and rOn-2000 ensembles for (a) the surface-based mixed layer depth averaged from the western domain edge to 50 km ahead of the algorithm-identified sea breeze front between 1200–1800 LT; and (b) the maximum sea breeze inland extent. The numbers on the horizontal axis refer to the arbitrarily identified simulation identities of each 130 member ensemble.



**Figure 6.** Cumulative frequency of low cloud (cloud top heights < 4 km AGL) occurrence in (a) rOn-500 and rOn-2000; (b) percentage differences of the low cloud frequency occurrence between rOn-500 and rOn-2000; (c) maximum cloud top height (km) over land between 1200–1800 LT; (d) percentage differences of the maximum cloud top height between rOn-500 and rOn-2000. Note that simulations with 100% low cloud frequency are not indicated in (a) for the sake of figure clarity; those columns without dots are therefore those simulations that only have clouds with cloud top heights less than 4 km AGL.







Figure 7. The (a) maximum, (b) mean, (c–f) 95, 75, 50, and 25th percentile of updrafts velocities greater than 1 m s$^{-1}$ over land between 1200–1800 LT in the rOn-500 (blue) and rOn-2000 (red) ensembles.



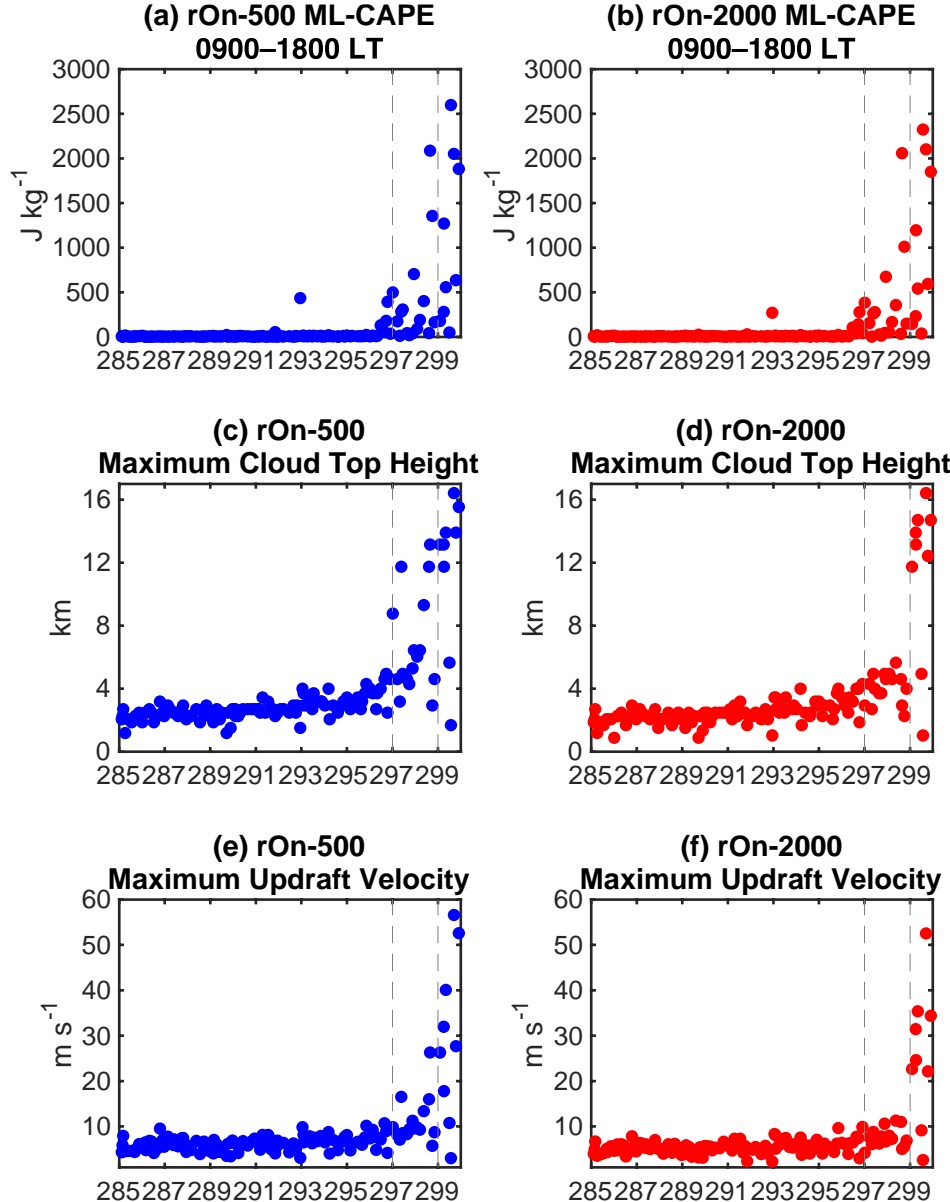

Initial Boundary Layer Potential Temperature (K)

**Figure 8. Pairwise scatterplots for the (a, b) spatially averaged (from 10 km ahead of the sea breeze front to the western domain edge) and temporally averaged (0900 - 1800 LT) mixed-layer CAPE (J kg$^{-1}$); (c, d) maximum cloud top height (km); (e, f) maximum updraft velocity (m s$^{-1}$) versus the initial boundary layer potential temperature (K) for the clean (left column) and polluted (right column) ensembles. The vertical dashed grey lines refer to the potential temperature thresholds described in the text.**

**Figure 9.** The percentage contribution to the variance of median updraft velocities in (a) rOn-500 and (b) rOn-2000, by each of the 10 environmental parameters of interest over the entire parameter space range (left stacked bar graphs), and then over the DRY, MID, WET soil regimes (second to fourth stacked bar graphs from the left). Mean responses of the median updraft velocities to the two-most important parameters, (c) soil saturation fraction and (d) inversion layer strength. Solid and dashed lines in (c) and (d) indicate the mean responses of the median updraft velocities of rOn-500 and rOn-2000, respectively. The numbers at the top of each plot in (c) and (d) indicate the percentage contribution of each parameter to the output variance. (e) Pairwise scatter plots of land-averaged surface sensible heat flux between 1200–1800 LT (W m⁻²) and soil saturation fraction. Blue and red colours indicate rOn-500 and rOn-2000, respectively. (f) is the same as (e) but for land-averaged latent heat flux (W m⁻²). Note that 130 data points in (e) and (f) have different values for the ten perturbed environmental parameters.







**Figure 10. (a)** Land-averaged accumulated surface precipitation at sunset (1800 LT) for the 130 ensemble members of rOn-500 (blue bars) and **(b)** rOn-2000 (red bars); and **(c)** percentage difference between rOn-500 and rOn-2000 for simulations that produce at least 0.1 mm of the land-averaged accumulated precipitation in rOn-500.





**Figure 11. Microphysical process rates (g kg⁻¹ (10 min)⁻¹) contributing to rain production. It should be noted that there is little difference in the rain gain and loss terms above 10 km, and so the y-axis is limited to enhance figure visibility. The rates are spatially and temporally averaged from 1200 LT to 1800 LT over the land regions. Gain terms are cloud to rain (solid), ice to rain (dashed), melting of ice (+ symbol), and vapor to rain (positive values of the dotted) processes. Loss terms are rain to ice (dot-dash) and rain to vapor (negative values of the dotted) processes. Only the 36 pairs that produce more than 0.1 mm of land-averaged surface accumulated precipitation by 1800 LT in the rOn-500 are shown here. Averaged freezing level between 1200–1800 LT for rOn-500 is marked by the grey line. Zero values are not shown for figure clarity. As such, values above the freezing level up to 10 km indicate that clouds exist up there.**





**Figure 12. A pairwise scatterplot of percentage difference of land-average accumulated precipitation at 1800 LT (mm) between rOn-500 and rOn-2000 versus initial boundary layer relative humidity (%). Only the 36 ensemble members that produce more than 0.1 mm in rOn-500 are included here. The colour scale indicates the initial boundary layer potential temperature. The size of each point is proportional to the initial soil saturation fraction. Numbers on top of four points represent simulation identification numbers.**