# Peer review of "The Modulation of Aerosol Impacts on Tropical Sea Breeze Systems by Low-Level Thermodynamic and Surface Conditions"

_Atmospheric Chemistry and Physics, 2021_

## Author Comment (AC1)

The authors would like to thank Dr. Matsui for his helpful and constructive suggestions. We have addressed each of his comments below. The reviewer comments are in black font, while the authors' responses are in blue for contrast. The * next to line numbers refers to the line number from the revised manuscript with track changes.

**Summary**:

This study conducted hundreds of sensitivity experiments of idealized cloud-resolving simulations in order to understand the effect of environmental parameters upon aerosol-sea-breeze convection interactions in tropics. Overall, set up of comprehensive sensitivity experiments, and statistical analysis (statistical emulation and variance-based analysis) are appealing aspects of this manuscript. However, the problem of this manuscript is that the figures are not summarizing and highlighting the physics very well. Although physics explanations are all reasonable, it's hard to extract essence from the figures. More specific major comments are described below. This paper has quite potential if revision goes well. So, I request "major revision" at this point.

**Major Comments:**

**Parameter ranges (Table 2**): I understand that some of them are derived from previous studies (Igel et al. 2018, Park et al. 2020). Are these perturbations in a realistic range? What do these perturbed ranges statistically mean? For example, soil saturation fraction between 0.1 and 0.9 are ranges from Savannah to tropical rainforest. Is this range typically happening in the real world of tropical coast regions? This question is also related to analysis of Fig 9a and 9b. When you compare different environmental factors, you should understand the natural ranges of these parameters, and should normalize/standardize them. Otherwise, you cannot state soil moisture has the largest impact on aerosol-cloud interactions.

Following one is my old paper that compared aerosol and thermodynamic impacts on low clouds by measuring 95%-frequency ranges of aerosol index and lower-tropospheric stability (Fig 3) in order to discuss relative importance.

Matsui, T., H. Masunaga, S.M.Kreidenweis, R.A.Pielke, Sr, W.-K. Tao, M. Chin, Y. Kaufman (2006), Satellite-based assessment of global warm cloud properties associated with aerosols, atmospheric stability, and diurnal cycle, *Journal of Geophysical Research– Aerosol and Clouds*. 111, D17204, doi:10.1029/2005JD006097.

You don't need to re-set new ranges of parameters for another hundreds of simulations, because you can just use a statistical emulator to estimate the relative impact of different parameters in standardized range. But, you have to understand statistical distributions of these parameters in the real world to understand "typical (one/two standard deviation)" ranges. With the standardized ranges of environmental parameters, you can state which parameters are important or not.

This is a great point. However, when we first started these studies, we could find very little in the literature regarding the maximum and minimum values of the parameters impacting sea breeze circulations, let alone their distributions. This is one of the reasons we performed these idealized tests across a wide range of values, as the emulator can then be used to select appropriate responses to appropriate parameter values. This said, we certainly strived to select a range of reasonable values, as described in Igel et al. (2018). We added the following statement.

> *Lines 117–121\*: The range in the selected variables were sourced from the sea breeze literature, the reasons for which were described in detail in Igel et al. (2018). While statistical distributions of the parameters of interest are very difficult to find in the literature, plausible parameter ranges for tropical regions were assigned to each of the variables tested. Should new observations provide greater constraints on the range of these parameters, the emulator approach (described below) allows for an assessment of the responses of the sea breeze convective system to the range of parameter values of interest.*

We added also the following sentence in section 5.2.2 regarding the soil moisture values and why we tested 0.1 to 0.9 soil saturation fraction.

> *Lines 385–390\*: It should be noted that sandy clay loam's observed soil saturation fraction varies from 0.25 to 0.75 along coastal equatorial Africa in June, July, and August (Rodell et al., 2004). Here we have extended the range tested to 0.1 through 0.9 to encompass slightly drier and wetter soil conditions in addition to those reported by Rodell et al. (2004) to take into account potentially more extreme conditions anticipated with changing climates. The corresponding variance-based sensitivity analyses for the daytime cumulus convection stratified by these three different soil regimes are shown in the second to fourth bar graphs from the left in Figures 9a and 9b.*

**Section 5.1 and 5.2 (Figures 5-7)**: I don't quite understand why you plot clean-polluted differences in zig-zag form, because simulation ID in X-axis does not represent physics at all. There should be a more effective way to represent this statistical representation. For example, histograms (clean, polluted, and clean-polluted) would be better to represent statistical differences, distributions, and significance of these sensitivity overall. Same issue also applies to Fig 10, too.

Thank you for this great suggestion. We initially plotted these figures as a function of simulation ID as in that way we could track the variability in aerosol cloud interactions as a function of each specific environment (identified by simulation ID) and hence address our question of environmental modulation of aerosol effects. However, we ultimately decided to remove the detailed environmental information and hence we agree that it no longer makes sense to plot these as a function of simulation ID. We have now revised Figures 5–7 and Figure 10 as you have suggested. These revised figures are shown below, along with the new text relevant to these figures.

In all of these figures, we constructed histograms for rOn-500 and rOn-2000, as well as the rOn-2000 minus rOn-500 differences, in order to emphasize aerosol-induced changes. For the histograms of rOn-2000 minus rOn-500, we added a zero vertical line to make it clear as to whether the aerosol impacts produced a positive or negative response. We also added blue (rOn-500) and red (rOn-2000) vertical lines on the histograms to show the ensemble-median value of corresponding characteristics. 
[revised manuscript text omitted]

**Section 5.2.1 (Figure 8):** You mentioned that "It is clear from this figure that….", but these scatter diagrams are not clear to me for comparison reasons. You may create a probability density grid scatter diagram (instead of dots), and you may plot clean-polluted. Or, at least, you may overlay scatter plots of clean and polluted like Fig 9e-f, and conduct some statistical process to mention "significant" or "clear" differences between clean and polluted cases.

Thank you for the suggestion. We tried a range of different graphical representations, including the probability density grid scatter diagram and the overlaid scatter plot, but found that the original scatter plots are the still the clearest representation of these trends. However, in order to better highlight the changes occurring between 297 and 299 K of the initial boundary layer potential temperature on the maximum updraft velocity, we added the 10 m s⁻¹ horizontal line to show that while 5 out of 12 deep convective simulations with the maximum updraft velocity greater than 10 m s⁻¹ in rOn-500 have initial boundary layer potential temperature between 297 and 299 K, the only simulations in rOn-2000 with the maximum updraft velocity significantly greater than 10 m s⁻¹ occur for initial boundary layer potential temperature greater than 299 K. We have also added the following text:

[Figure]

**Figure 8. Pairwise scatterplots for the maximum updraft velocity (m s⁻¹) versus the initial boundary layer potential temperature (K) for the (a) rOn-500 and (b) rOn-2000 ensembles. The vertical dashed grey lines refer to the potential temperature thresholds described in the text.**

*Lines 354–362\*: Figure 8 shows scatter plots relating the maximum updraft velocity to the initial boundary layer potential temperature for all 130 ensemble members in rOn-500 (Figure 8a) and rOn-2000 (Figure 8b). It is clear from this figure that the sea breeze-initiated deep convective mode (updrafts are greater than 10 m s⁻¹) in rOn-500 occurs in all of the ensemble members in which the initial boundary layer potential temperature is 297 K or greater, and in which the mixed layer CAPE is greatest (not shown). However, in rOn-2000 the threshold above which the deep convective mode occurs is 299 K, which is 2 K greater than that in rOn-500. For instance, in rOn-500 while 5 out of 12 deep convective simulations with the maximum updraft velocity greater than 10 m s⁻¹ have initial boundary layer potential temperatures of between 297 and 299 K (Figure 8a), the*

*only simulations in rOn-2000 with the maximum updraft velocity greater than 10 m s$^{-1}$ occur for initial boundary layer potential temperatures greater than 299 K (Figure 8b).*

**Fig 11**: Fig 11 does not summarize physics very well. It pretty much displays all cases. For example, if you compute clean-polluted differences in auto-conversion profiles, and you can create CFAD to summarize all cases in one plot for each microphysical process (melting of ice, ice-to-rain, rain-to-ice, cloud-to-rain, etc..), it would be nice, because Test ID does not show any information of environmental factors anyway. So far, it's too numerous and mechanical test ID. So, it's difficult to extract physics from this plot.

Thank you for the suggestion. As discussed above, our initial reasoning for doing this was in an attempt to keep track of the different environments producing the range in responses. However, given that we ultimately decided to eliminate tracking each of the specific environments, the original Figure 11 was less meaningful. We tried CFADs for each term, however, we found that these were not overly helpful in conveying our message. We ultimately wanted to clearly demonstrate that the 36 pairs with 36 different initial conditions all exhibit aerosol-induced decreases in cloud-to-rain and rain evaporation rates regardless of their different environments. As such, we decided to summarize rOn-2000 minus rOn-500 averaged process rates as shown below in Figure 11, in which all 36 pairs are plotted together. We also plotted the ensemble mean rOn-2000 minus rOn-500 values for cloud-to-rain and rain-to-vapor plots with thick gray lines, as all of the 36 pairs exhibit warm rain processes. While the cold rain processes (i.e., melting of ice, rain to ice, and ice to rain process) contributing to surface precipitation are included in Figure 11, a detailed discussion and representation of cold rain processes are not included in the revised manuscript, as not all 36 pairs exhibit cold rain processes.

*Lines 442–455\*: The bulk microphysical processes contributing to the differences in the surface rainfall in the 36 precipitating ensemble pairs are shown in Figure 11. All of the following source and sink terms for rain are considered:*

1. *Cloud-to-rain: cloud water transferred to rain through collection (gain term; Figure 11a);*

2. *Rain-to-vapor: evaporation of liquid water from rain (loss term; Figure 11b);*

3. *Melting of ice: ice mass transferred to rain via thermodynamic melting as the ice species fall below the freezing level (gain term; Figure 11c);*
4. *Rain-to-ice: rainwater that is collected by ice species through riming (loss term; Figure 11d); and*
5. *Ice-to-rain: collisional ice melting due to collection of warmer rain (gain term; Figure 11e).*

*These process rates are averaged across all grid points over the land domain between 1200 and 1800 LT. In most of the precipitating members, the cloud frequency over land is*

*heavily weighted by the daytime cumulus convection mode. As a result, the frequency and associated contributions made by the averaged mixed-phased process contributions are small, if even existent. However, in some members, as shown in Figures 11c–e, the averaged mixed-phase process contributions are greater than those of the warm-phase processes.*

*Lines 459–463\*: Similarly, average rain evaporation rates (i.e., rain-to-vapor) shown in Figure 11b are also greater in magnitude in rOn-500 than rOn-2000, implying that the population of less numerous but larger raindrops formed in rOn-2000 evaporate less readily. The production of populations of fewer but larger raindrops in polluted conditions has been observed previously (e.g., Altaratz et al. 2007; Storer and van den Heever 2013).*

*Lines 466–469\*: Figures 11c–e show that enhanced aerosol loading primarily produces a reduction in all three of the cold rain processes contributing to the rain budget, with only minor increases in the profile with aerosol loading in some cases. However, given the small sample size, additional testing would be required before conclusive statements can be made regarding environmental modulation of cold phase processes.*

[Figure]

**Figure 11. Aerosol-induced differences to the processes generating rain for the range of environmental conditions tested in these large ensemble experiments. Shown are the rOn-2000 minus rOn500 differences (a) cloud to rain; (b) rain to vapor; (c) melting of ice; (d) rain to ice; and (e) ice to rain rates (see the text for an explanation of these processes) using the 36 ensemble pairs that produce at least 0.1 mm of land-averaged surface accumulated precipitation in rOn-500. Process rates are averaged over the land domain between 1200 and 1800 LT. The thin black lines in all of the figures are from all 36 pairs, and the thick grey lines in a and b are the means of the rOn-2000 minus rOn-500 for the 36 pairs.**

**Minor Comments:**

Resolution: Simulations are conducted with 1km grid spacing, and discussion of shallow-to-deep convection transition can be limited. I understand this is purely because of computational limitations with the many ensemble simulations. At least, you should mention this limitation somewhere in the manuscript.

We have added the following sentence in the revised manuscript.

> *Lines 537–540\*: Thirdly, a grid spacing of 1 km was selected for these extensive ensembles given their high computational costs. Such a grid spacing will marginally resolve deep convective cloud systems but will under resolve the shallow convective mode. As computational capabilities are enhanced, a similar study should be conducted using grid spacings of O(100m).*

Line 37: I suggest ditch following sentence of this paper's topic "*Such organised tropical convection also plays an essential role in global climates via its impacts on planetary circulations such as the Walker circulation or the Madden-Julian Oscillation (Hendon and Woodberry, 1993; Zhang, 2005).*" This paper is not dealing with organized tropical convection.

It has been removed.

Line 63: "convectively" -> "convective"

It has been corrected.

Line 68: Suggest ditch "in the interest of focusing specifically on aerosol indirect effects". Sounds repetitive.

Done.

Line 70: "size and composition" -> "sizes and compositions"

Done.

Line 88: "theories" -> "hypothesis" Also apply the following sentences.

Thank you, we have made the change.

Line 139: Table 1 is not refered from sentences.

Table 1 was referred to in Section 2.1., Line 128.

Line 243: Add "and less surface turbulent heat flux" after "With less surface upwelling longwave radiation,".

Done.

Line 245-246: Remove parenthesis.

It has been removed.

Line 247: "longwave radiation" -> "longwave radiation and surface turbulent heat flux"

Done.

Figures 3 and 4 (and related discussion) might be combined, since these are all surface impact and feedback.

Thank you for the suggestion. After considering this point, we have decided to keep Figures 3 and 4 as they are since Figure 3 is for showing land vs. ocean surface radiation and temperature responses, whereas Figure 4 is only for land surface heat fluxes.

---

## Author Comment (AC2)

The authors would like to thank Referee 2 for their constructive and helpful suggestions. We have addressed each of their comments below. The reviewer comments are in black font, while the authors' responses are in blue for contrast. The * next to line numbers refers to the line number from the revised manuscript with track changes. The * next to figure number indicates that the figure is only included in this response to the reviewer document, not in the revised manuscript.

Overview

This study performs 2 large, idealized simulation ensembles of sea breeze convection evolution covering a range of atmospheric and surface parameters, 1 with low aerosol loading and the other with high aerosol loading, with a statistical emulator used to fill in sensitivities across a greater range of conditions. The authors find that increased aerosol loading via reduction of incoming shortwave radiation inhibits surface fluxes and the land-sea thermal contrast that drives the daytime sea breeze. This thus contributes to weaker convection along the sea breeze front, particularly for warm clouds and through suppression of deep convection initiation. Once deep convection initiates, aerosol effects on convective updrafts are modulated by other atmospheric conditions. Under all conditions, increased aerosol loading suppresses precipitation with a magnitude that is modulated by other atmospheric conditions. It is further found that soil saturation fraction is the most important modulator of updraft velocity variance in cumulus clouds ahead of the sea breeze of the parameters tested.

Overall, this is a well written study, although it could be more succinct in some places that I note in the detailed comments. Although the importance of aerosol direct effects on atmospheric thermodynamic structure and clouds is known, it has largely been ignored in recent studies focusing on microphysical impacts from aerosols. This study connects these two pieces over a wide range of low-level thermodynamic states, providing a nice addition to literature in this area. I have several "major" concerns regarding choices and interpretations of some analyses. In addition, analyses mix shallow and deep convection together, but they are quite different dynamically and microphysically (e.g., shallow clouds not necessarily being buoyancy driven and not containing ice) and thus have some differences in key environmental influences on them. Therefore, grouping them together is confusing and perhaps also misleading in terms of several interpretations of analyses. Lastly, more caveats should be discussed such that results can be more appropriately interpreted within the context of real-world convective clouds. More details are provided in the specific comments below. All should be mostly straightforward to address.

**Major Comments**

1. More specific language can be used in places that avoids over-generalized conclusions from the analyses. In particular, the several parameters varied are limited to the surface through boundary layer top inversion layer. However, as mentioned in the introduction, there can be substantial sensitivities to vertical wind shear and free tropospheric RH, so the results cannot be generalized to sea breeze environments in general. In addition,

there is no microscale and mesoscale variability in surface conditions, thermodynamics, or circulations that are environmental conditions affecting cloud evolution, but these not considered in the idealized setup. Of course, this is perfectly fine and not everything can be covered in a single study, but then the title and results would be more accurate by referring to low level thermodynamic impacts rather than environmental impacts in general.

This is a good point. We have changed the title to *"The Modulation of Aerosol Impacts on Tropical Sea Breeze Systems by Low-Level Thermodynamic and Surface Conditions"* and throughout the manuscript, we have tried to better clarify what we mean by "environment" as follows:

> *Lines 115–117\*: As described in Park et al. (2020), we make use of 130 different initial environmental conditions, where ten different lower-tropospheric thermodynamic, wind, and surface properties (Table 2) are simultaneously perturbed across a range of values representative of tropical equatorial regions.*
>
> *Lines 475–477\*: Each of these two ensembles was comprised of 130 members initialised with 130 different initial conditions, representing the simultaneous perturbation of ten thermodynamic, wind, and surface properties, the ranges of which were sourced where possible from the current sea breeze literature.*

2. More caveats to better contextualize conclusions are needed. Here are some examples:
   1. The horizontal grid spacing at 1 km is too coarse to resolve the primary convective updrafts in deep convection, let alone cumulus clouds.

      We have added the following sentences to the conclusion of the revised manuscript.

      > *Lines 537–540\*: Thirdly, a grid spacing of 1 km was selected for these extensive ensembles given their high computational costs. Such a grid spacing will marginally resolve deep convective cloud systems but will under resolve the shallow convective mode. As computational capabilities are enhanced, a similar study should be conducted using grid spacings of O(100m).*

   2. RH is always relatively high in the boundary layer. Does this contribute to updrafts intensifying as SHF increases and LHF decreases? In other words, are there potentially conditions in which increasing LHF with equal decreases in SHF will increase updrafts?

      We are not exactly sure what the reviewer is asking here and request further clarification. The reviewer first asks whether given the relatively high RH in the boundary layer as to whether this contributes updrafts intensifying as SHF

*increases* and LHF *decreases*, but goes on to refer to *increasing* LHF with equal *decreases* in SHF. If the reviewer is asking as to whether the fact that the relatively moist boundary layer allows for updraft intensification with enhanced SHFs, even though the LHF decreases, then yes that is the case. The moisture already within the BL plays a role in offsetting the loss of water vapor due to reduced LHFs.

3. Some past studies have shown highly non-linear effects of aerosols on clouds, but that cannot be assessed with only 2 different aerosol concentrations. How does that affect the robustness of conclusions?

The reviewer is quite correct that our study cannot encompass the entire spectrum of aerosol-cloud responses. We would have loved to include an additional suite of moderate aerosol loading ensembles but this became computationally unfeasible. We have however added an acknowledgement of this and some additional considerations for future work in the conclusions.

*Lines 533–537\*: Secondly, only two different aerosol loadings were examined in this study. Some modelling studies have reported non-monotonic responses of convective clouds and precipitation to enhanced aerosol loading (Storer and van den Heever, 2013; Dagan, 2017; Liu et al., 2021). As such, a future study should assess the potential non-linear effects of aerosol on convection across a wide range of surface and meteorological environments by conducting ensembles with a broader range of aerosol loadings.*

3. The discussion around the 3 ingredients for moist convection is confusing since they apply to deep convection rather than shallow convection and the plots are not actually of the 3 ingredients but of variables that are related in some way.
   1. First, the changes in moisture, instability, and lift can be more clearly shown. For example, why not show boundary layer mixing ratio differences ahead of the sea breeze line?

   Thank you for these helpful comments. The reviewer makes a good point that the three convection ingredients may confuse readers because Doswell's ingredients only apply to deep convective systems. Therefore, we removed the "ingredient" terminology from the manuscript. Instead, we have simply discussed those processes important to driving the shallow and deep convective modes including: (1) latent heat flux changes reflecting reduced moisture via evapotranspiration, evaporation, and condensation (Figures 5a and b), (2) boundary layer instability represented by mixed layer depth (Figures 5c and d), (3) convective instability for deep convection presented by mixed-layer CAPE (Figures 5e and f), (4) sea breeze convergence shown by the maximum inland extent (Figures 5g and h), and maximum updraft velocities at the vicinity of the algorithm-identified sea breeze front (Figures 5i and j).

.

[Figure]

**Figure 5.** Histograms of (a) the land-surface latent heat flux (W m⁻²), (c) the surface-based mixed layer depth (km), and (e) the mixed-layer CAPE (kJ kg⁻¹), all of which are averaged for each ensemble from the western domain edge to 50 km ahead of the algorithm-identified sea breeze front between 1200–1800 LT; (g) the maximum inland extent of the sea breeze front (km); and (i) maximum updraft velocities at the algorithm-identified sea breeze front±1 km (m s⁻¹). The median values of each characteristic are marked by the red and blue thin vertical lines. The light blue shading and the red lines represent rOn-500 and rOn-2000 ensembles, respectively. Figures (b, d, f, h, and j) are histograms of the differences in the corresponding fields shown in (a, c, e, g, and i) arising due to aerosol loading (rOn-2000 minus rOn-500). The dashed magenta lines indicate where the difference between rOn-2000 and rOn-500 is zero. The bin width of each histogram is marked at the upper corner of each panel.

2.  For instability, Doswell (1987) explicitly refers to "conditional instability" in relation to deep convection. This is not a function of boundary layer depth as is implied but instead a function of free tropospheric lapse rates, as is stated in Doswell (1987). Therefore, a better measure of instability is CAPE than boundary layer depth. Granted, this also assumes that the cumulus clouds forming are buoyancy driven rather than mechanically driven (e.g., intrusion of a saturated rising boundary layer thermal into an inversion layer), which is likely not the case in most scenarios based on Figure 8. In addition, Doswell (1987) discusses deep convection and says nothing of ingredients for shallow convection. The mixture of these 2 types of clouds with many of the setups having no CAPE such that shallow clouds are simply an extension of boundary layer dry thermals rather than buoyant clouds like deep convection leads to some confusing messaging.

As stated above, in order to avoid confusion, we removed the "ingredient" terminology. We have also added the distribution of the average mixed layer CAPE ahead of the sea breeze in Figure 5 (Figure 5e). We kept the histograms of mixed layer depth (Figure 5c) since they are still a useful indicator of instability in the boundary layer and they are important to the depths of shallow cumulus. While we agree with the reviewer that examining aerosol impacts on both of the modes comprising the sea breeze systems can at times be confusing, as our goal was to examine the impacts of aerosols on the sea breeze system as a whole, we felt it was necessary to present the results in this way. We have, however, strived to better enhance the clarity of these results, first by removing the "ingredient" terminology as stated above, and also by trying to make the modal discussions clearer where we can.

*Lines 242–254\*: These reductions in sensible and latent heating will negatively impact the convective boundary layer by limiting both the heating and moistening of this layer Specifically, less moisture will be available for convection via evapotranspiration, evaporation, and condensation, which is reflected in the reduction of the ensemble-median surface latent heat flux in the polluted case (Figure 5a), as well as the differences between the polluted and clean ensembles, by far the majority of which are negative (Figure 5b). The surface-based mixed layer depth, defined here as the level above the surface at which the vertical gradient of the potential temperature first exceeds 2 K km$^{-1}$, decreases in rOn-2000 compared with rOn-500 due to this reduction in surface sensible heat flux and associated turbulent mixing. Histograms of the mean surface based-mixed layer depth ahead of the sea breeze front in rOn-500 and rOn-2000 and their differences are shown in Figures 5c and 5d, respectively. In Figure 5c, the mean mixed layer depth distribution shifts toward lower values with the change from rOn-500 to rOn-2000. This aerosol-induced*

*decrease in mixed layer depth is also evident in the reduced ensemble-median values (vertical lines in Figure 5c) in rOn-2000 compared with rOn-500. Figure 5d further indicates that the mixed layer in each member of the rOn-2000 ensemble is shallower than the corresponding mixed layer in the rOn-500 ensemble, with all of the rOn-2000 minus and rOn-500 values being negative.*

*Lines 255–266\*: While mixed layer depth is a valuable indicator of instability in the boundary layer and hence the depths of shallow cumulus, CAPE is a more pertinent assessment of instability for the deep convective clouds driven by the sea breeze convergence. As shown in Figure 5e, most of the simulations in both ensembles have averaged mixed-layer CAPE values close to zero which is in keeping with the fact that only a handful of the ensemble members produce deep convection. The ensemble-median values are slightly reduced with enhanced aerosol loading, from 7.6 to 7.1 J kg$^{-1}$. Figure 5f also demonstrates that the differences in CAPE between rOn-500 and rOn-2000 may be positive or negative but are mostly quite small in magnitude. The exceptions to this are the magnitudes for those cases that produce deep convection, where the CAPE values may be as high as 2102 J kg$^{-1}$ and exceed marginal CAPE. The aerosol-induced differences are all negative and range in magnitude from 8 to 115 J kg$^{-1}$, except for one ensemble pair with a positive difference of 50 J kg$^{-1}$. Therefore, while the variations in CAPE with aerosol loading appear to be small in magnitude for most members of the ensembles, they may play a discriminating role in aerosol impacts on deep convective updraft velocities for those cases that do support deep convection. This is discussed further in section 5.2.1.*

3. And then for lift, a strong sea breeze will push further inland, but this doesn't say anything about the depth of the sea breeze, which may be more relevant to the lift at the cloud level such that the cloud can penetrate through a deeper layer. I wonder if this can be greater in weaker sea breeze conditions if the sea breeze convergence is deeper from the front being less sloped and better balanced with zonal winds. Would it be better to examine vertical motion along the front?

The reviewer makes a great point. Both horizontal and vertical length scales of the sea breeze circulation have been used to present sea breeze intensity in the literature (Crosman and Horel, 2010). To quantitatively represent the vertical length scale of the sea breeze front every point in time for each point along the front for all of the simulations, one would need to utilize a complete 3D tracking algorithm. However, we have calculated histograms of the distribution of the maximum updraft velocities in the region 1 km ahead of and behind the algorithm-identified sea breeze front to assess the contributions by the sea breeze front to lift. We examined the maximum velocity 1km ahead of and

behind the surface identified front to account for any forward bulging or backward tilting of the frontal boundary in relation to the surface identified location of the front. We have kept the sea breeze extent as this is another measure of the sea breeze intensity and associated lift.

*Lines 266–285\*: We now turn our attention to the vertical lift provided by the convergence along the sea breeze front in all of the simulations. Classical sea breeze theory dictates that, to first order, the faster the sea breeze moves, the further inland the sea breeze travels during the day, the stronger the convergence along the sea breeze front, and hence the greater the vertical lift along the front. Here we examine the maximum inland extent of the sea breeze front, as well as the maximum updraft velocity found within $\pm$ 1 km of the algorithm identified surface location of sea breeze front during the afternoon (1200 LT–1800 LT). The maximum inland extent of the sea breeze front is identified as the last inland location of the sea breeze front detected by the sea breeze front algorithm (Igel et al., 2018). We assess the maximum vertical velocity within $\pm$ 1 km of the surface location of the front in order to account for any forward bulging or backward tilting of the frontal boundary in relation to the identified location of the front at the surface. The distribution of the maximum sea breeze inland extent shows a shift towards lower values with enhanced aerosol loading (Figure 5g), also evident in the decrease in the ensemble median with enhanced aerosol loading (Figure 5g). It is also evident from Figure 5h that the sea breeze extent is less in rOn-2000 than rOn-500 for each and every one of the ensemble pairs, thus demonstrating the significant role of aerosol loading and the direct effect on this baroclinic circulation, and the subsequent forcing of deep convection. The distribution of the maximum updraft velocities (and hence lift) along the sea breeze front shows a shift towards reduced updraft velocities in more polluted environments, as is demonstrated by the small reduction in the ensemble median of rOn-2000 compared with rOn-500. However, in spite of the robust response of the maximum inland extent of the sea breeze to aerosol loading (Figure 5h), the impacts of enhanced aerosol loading on the maximum frontal velocities do not always produce a negative vertical velocity response (Figure 5j). This suggests that while the environment does not appear to modulate the direct impacts of aerosols on the sea breeze dynamics and inland extent, it may locally modulate aerosol impacts on the updraft velocities, possibly through aerosol indirect processes and/or changes to CAPE.*

4. Some figures can be improved.

1. It's not clear what Figure 6a is showing. Is it the percentage of cloudy columns that have cloud tops < 4 km AGL? In other words, is this ignoring non-cloudy columns such that it is not a cloud fraction? Also, why not have the y-axis extend down to 0% so that the fraction in all simulations can be quantified? The statement on lines 317-318 that most simulations only have low clouds isn't clear from Figure 6a-b in which most simulations don't even appear to have a percentage value or difference between ensembles. In addition, Figure 6b isn't referred to in the text, so is it needed?

In response to suggestions from Reviewer 1, we have revised Figure 6 by constructing histograms for rOn-500 and rOn-2000 (left column in Figure 6), as well as the differences between rOn-2000 and rOn-500 (right column in Figure 6), in order to emphasize aerosol-induced changes. For the histograms of rOn-2000 minus rOn-500, we added a zero vertical line to clarify whether the aerosol impacts produced a positive or negative response. We also added blue (rOn-500) and red (rOn-2000) vertical lines on the

[Figure]

Figure 6. Similar to Figure 5 but for histograms of the (a) low cloud (cloud top height < 4 km) fractional contribution to the total number of cloudy columns (low cloud columns / all cloudy columns) and (c) the maximum cloud top height. The right column (b and d) represents rOn-2000 minus rOn-500 difference histograms for the corresponding fields (a and c) in the left column. The dashed magenta lines indicate where the difference between rOn-2000 and rOn-500 is zero. The median values of each characteristic are marked with vertical lines in the left column. The bin width of each histogram is marked at the upper corner of each panel.

histograms to show the ensemble-median value of corresponding characteristics. The bin width of each histogram is marked on each panel.

We have also strived to better clarify the low cloud contribution. In response to the reviewer's question, yes, we consider cloudy columns to calculate frequency occurrence. We have included this in the revised version of the manuscript as follows:

*Lines 300–309\*: We examine the impacts of aerosol loading on cloud top height in two ways. First, we examine the frequency distribution of the fraction of low (cloud top height < 4 km) cloudy columns to the total number of cloudy columns for all simulations (low cloud columns / all cloudy columns) (Figure 6a). Among the 130 simulations in each ensemble, there are 104 and 113 simulations with low clouds only in rOn-500 and rOn-2000, respectively. The vast majority of ensemble members in both ensembles are therefore dominated by low clouds, as demonstrated by the ensemble-median values of 100% (Figure 6a). In other words, only shallow convective clouds develop both ahead of and along the sea breeze front in most of the ensemble members, and while all of the environmental conditions tested here support the development of the sea breeze-initiated shallow convective mode, most do not support the development of the sea breeze-initiated deep convective mode. The difference between the two ensembles (Figure 6b) shows that in the majority of the simulations, the low cloud fraction stays the same or is weakly increased with enhanced aerosol loading.*

2. For Figure 7, are updrafts any grid points > 1 m/s or is a condensate constraint applied to ensure that they are cloudy updrafts? It would make sense to confine these to clouds given the focus of the paper. It's also not necessary to show some panels like the 25th, 75th and 95th percentiles that don't provide any additional information beyond what can be concluded in the other panels.

[Figure]

**Figure 7. Similar to Figure 5 but for (a) histograms of the maximum updraft velocity in rOn-500 and rOn-2000 and (b) a histogram of the maximum updraft velocity differences arising from aerosol loading (rOn-2000 minus rOn-500).**

In response to Reviewer 1's suggestions, we have replaced Figure 7 with the histogram as shown above. Figure 7 contains histograms for maximum updraft velocity in rOn-500 and rOn-2000, and the rOn-2000 minus rOn-500 to highlight aerosol-induced changes. For the histogram of rOn-2000 minus rOn-500, we included a zero vertical line to make it clear as to whether the aerosol impacts produced a positive or negative response. Blue (rOn-500) and red (rOn-2000) vertical lines on the histogram show the ensemble-median value of corresponding characteristics. The bin width of each histogram is marked on each panel. We have included the following text in association with Figure 7.

*Lines 334–338\*: Overall aerosol-induced suppression of the maximum updraft velocities is evident by the decrease in the ensemble-median values of the maximum updraft velocities (Figure 7a). However, Figure 7b implies that enhanced aerosol loading may produce either weaker or stronger updraft velocities depending on the initial environmental conditions and suggests that such aerosol-induced responses are environmentally modulated.*

In response to the reviewer's question regarding constraining the results by condensate, we did not include such a threshold, and simply chose maximum vertical velocity values from those points where the updrafts are greater than 1 m s$^{-1}$. We used this approach as we wanted to include boundary-layer dry thermals that are driven by surface heating ahead of the sea breeze front. Such thermals may not have yet developed condensate. However, if you compare the maximum updrafts with and without the condensate constraint (0.05 g kg$^{-1}$), as shown in Figure 1*, the r values were 0.9986 and 0.9989 for rOn-500 and rOn-2000, respectively, and thus the differences between using and not using the condensate threshold are small.

[Figure]

**Figure 1\*. Scatter plots of the maximum updraft velocity without condensate constraint (x-axis) and with condensate constraint of 0.05 g kg$^{-1}$ (y-axis) for (left) rOn-500 and (right) rOn-2000 ensemble. The gray line is a one-to-one line.**

3. Figure 11: I suggest brainstorming a way to reduce the panels in this figure in some summarizing way that supports the primary conclusions summarized in the text. A reader cannot be expected to comprehend the takeaway messages from 36 panels with 8 different lines and 2 different symbols nor will they likely try.

Yes, we agree. Our initial reasoning for doing this was to keep track of the different environments producing the range in responses, which was why we had plotted these as a function of simulation ID. However, we ultimately decided to eliminate tracking the environments, and hence the original Figure 11 became less meaningful. We have now replotted Figure 11 as process differences for all 36 pairs, as shown below. The black lines represent each member of the 36 pairs (as a difference) and the ensemble means of rOn-2000 minus rOn-500 are shown with thick gray lines. While the cold rain processes (i.e., melting of ice, rain to ice, and ice to rain process) contributing to surface precipitation are included in Figure 11, a detailed discussion on cold rain processes is no longer included in the revised manuscript, as not all 36 pairs exhibit cold phase processes, and it became challenging to meaningfully represent these differences using one panel.

[Figure]

**Figure 11. Aerosol-induced differences to the processes generating rain for the range of environmental conditions tested in these large ensemble experiments. Shown are the rOn-2000 minus rOn500 differences (a) cloud to rain; (b) rain to vapor; (c) melting of ice; (d) rain to ice; and (e) ice to rain rates (see the text for an explanation of these processes) using the 36 ensemble pairs that produce at least 0.1 mm of land-averaged surface accumulated precipitation in rOn-500. Process rates are averaged over the land domain between 1200 and 1800 LT. The thin black lines in all of the figures are from all 36 pairs, and the thick grey lines in a and b are the means of the rOn-2000 minus rOn-500 for the 36 pairs.**

*Lines 442–455\*: The bulk microphysical processes contributing to the differences in the surface rainfall in the 36 precipitating ensemble pairs are shown in Figure 11. All of the following source and sink terms for rain are considered:*

1.  *Cloud-to-rain: cloud water transferred to rain through collection (gain term; Figure 11a);*

2.  *Rain-to-vapor: evaporation of liquid water from rain (loss term; Figure 11b);*

3.  *Melting of ice: ice mass transferred to rain via thermodynamic melting as the ice species fall below the freezing level (gain term; Figure 11c);*
4.  *Rain-to-ice: rainwater that is collected by ice species through riming (loss term; Figure 11d); and*
5.  *Ice-to-rain: collisional ice melting due to collection of warmer rain (gain term; Figure 11e).*

*These process rates are averaged across all grid points over the land domain between 1200 and 1800 LT. In most of the precipitating members, the cloud frequency over land is heavily weighted by the daytime cumulus convection mode. As a result, the frequency and associated contributions made by the averaged mixed-phased process contributions are small, if even existent. However, in some members, as shown in Figures 11c–e, the averaged mixed-phase process contributions are greater than those of the warm-phase processes.*

*Lines 459–463\*: Similarly, average rain evaporation rates (i.e., rain-to-vapor) shown in Figure 11b are also greater in magnitude in rOn-500 than rOn-2000, implying that the population of less numerous but larger raindrops formed in rOn-2000 evaporate less readily. The production of populations of fewer but larger raindrops in polluted conditions has been observed previously (e.g., Altaratz et al. 2007; Storer and van den Heever 2013).*

*Lines 527–529\*: Similarly, average rain evaporation rates (i.e., rain-to-vapor) shown in Figure 11b are also greater in magnitude in rOn-500 than rOn-2000, implying that the population of less numerous but larger raindrops formed in rOn-2000 evaporate less readily.*

*Lines 466–469\*: Figures 11c–e show that enhanced aerosol loading primarily produces a reduction in all three of the cold rain processes contributing to the rain budget, with only minor increases in the profile with aerosol loading in some cases. However, given the small sample size, additional testing would be required before conclusive statements can be made regarding environmental modulation of cold phase processes.*

5. Some conclusions are questionable, and at the very least, should be softened.
    1. Lines 474-475: I'm not sure it can be concluded that the environment modulates the cold rain response to aerosol loading with so few ensemble members producing ice. If you were to impose some white noise thermodynamic perturbations, would similar effects in magnitude be seen? In other words, are differences for any 1 ensemble member robust? That isn't shown, so I think the most that can be claimed is that the responses of deep convection to aerosol concentration across different low level thermodynamic conditions are uncertain rather than claiming that they are robustly modulated by environment.

        Yes, this is a good point. We have revised the precipitation section, only focusing on the warm rain process while both warm and cold rain processes are included in Figure 11.  Due to the limited number of samples, cold rain processes are not further analyzed. As the reviewer pointed out, there are only a few cold rain cases, while all of the 36 pairs contain warm rain responses.

        *Lines 466–469\*: Figures 11c–e show that enhanced aerosol loading primarily produces a reduction in all three of the cold rain processes contributing to the rain budget, with only minor increases with aerosol loading in some cases. However, given the small sample size, additional testing would be required before conclusive statements can be made regarding environmental modulation of cold phase processes.*

        *Lines 522–524\*: Cold rain processes also showed an overall aerosol-induced reduction, however, given the small sample size, the robustness of these trends is uncertain and would require additional experiments before more definitive statements can be made.*

    2. Comparing tests 75 and 110 is really comparing apples to oranges, so to speak, and is an example of confusing messaging due to mixing deep convection with shallow convection. Test 75 is a deep convective case with a lot of cold phase precipitation. Because cold phase precipitation will form in this case regardless of aerosol concentration, the aerosol effect on precipitation may be reduced relative to a case with only shallow, warm clouds like test 110 because the CCN concentration directly affects liquid hydrometeors but indirectly affects ice hydrometeors. With conditions being suitable for deep convection in one test and not in the other, the sensitivities of precipitation to surface or boundary layer thermodynamic conditions cannot be expected to be robust or interpretable. Because of this, deep convection with ice and shallow convection without ice should probably be separated in analyses for easier interpretation (which may help with connections to the 3 ingredients discussion as well that becomes confusing with shallow and deep clouds combined). In addition, it seems clear that increase boundary layer temperature decreases precipitation differences in Figure 12, which is likely related to liquid water path increases for the shallow convection, but the sensitivity to RH and saturation

fraction are lost on me in this figure such that they don't seem robust, and I'm not sure the discussion of select cases in the text matters if there isn't a robust signal that can be seen.

> We agree with the reviewer that this is not a fair comparison of apples to oranges and have removed Figure 12. Lines 476–499 have also been deleted.

**Minor Comments**

1. The manuscript could benefit by cutting out extraneous text that is repetitious and/or distracting from the primary messages. Consider shortening the abstract and conclusions to better highlight the primary messages. In addition, consider removing the summaries in the results sub-sections.

   We have revised the abstract as follows:

   > *Lines 7–27\*: This study investigates how enhanced loading of microphysically and radiatively active aerosol particles impacts tropical sea breeze convective systems and whether these impacts are modulated by the many environments that support these cloud systems. Comparisons of two 130-member clean and polluted ensembles demonstrate that aerosol direct effects reduce the surface incoming shortwave radiation and the surface outgoing longwave radiation. Changes in ensemble median values of the surface latent heat flux, mixed layer depth, mixed layer convective available potential energy, maximum inland sea breeze extent and the sea breeze frontal lift suggest that enhanced aerosol loading generally creates a less favourable environment for sea breeze convective systems. However, the sign and magnitude of these aerosol-induced changes are occasionally modulated by the surface, wind, and low-level thermodynamic conditions. As reduced surface fluxes and instability inhibit the convective boundary layer development, updraft velocities of the daytime cumulus convection developing ahead of the sea breeze front are robustly reduced in the polluted environments across the environments tested. Statistical emulators and variance-based sensitivity analyses reveal that the soil saturation fraction is the most important environmental factor contributing to the updraft velocity variance of this daytime cumulus mode, but that it becomes a less important contributor with enhanced aerosol loading. It is also demonstrated that increased aerosol loading generally results in a weakening of the sea breeze-initiated convection. This suppression is particularly robust when the sea breeze-initiated convection is shallower, and hence restricted to warm rain processes. While the less favourable convective environment arising from aerosol direct effects also restricts the development of sea breeze-initiated deep convection in some cases, the response does appear to be environmentally modulated, with some ensemble members producing stronger convective updrafts in more polluted environments. Sea breeze precipitation is ubiquitously suppressed with*

*enhanced aerosol loading across all of the environments tested, however, the magnitude of this suppression remains a function of the initial environment. Altogether, our results highlight the importance of evaluating both direct and indirect aerosol effects on convective systems under the wide range of convective environments that support their development.*

We have also strived to reduce repetition wherever possible, including removing sub-section summaries such as those shown below.

*Lines 296–298: In the previous section, we demonstrated that the three convective ingredients of moisture, instability, and lift, are reduced in the polluted ensemble, when compared with the pristine ensemble, and that this is due to the direct impacts of enhanced aerosol loading on incoming solar radiation.*

*Lines 436–440: To summarize, the daytime cumulus updraft velocities become weaker with enhanced aerosol loading. This is consistent with the reduction in surface fluxes and mixed layer depth under polluted conditions described in section 4. The ten-dimensional variance-based sensitivity analysis described here demonstrates that the relative importance of the soil saturation fraction and inversion layer strength in contributing to the updraft velocity variance varies as a function of aerosol loading. The degree of aerosol-induced weakening of the updrafts is thus modulated by the aerosol-environmental feedbacks.*

2. Lines 79-81: There are also observation (e.g., Varble 2018) and modeling (Grabowski and Morrison 2016, 2020) that show opposing conclusions and should be cited somewhere (e.g., on line 90).

This was poor oversight on our behalf. We agree and have included those references here:

*Lines 76–80\*: These warm-phase aerosol-cloud dynamical feedbacks have collectively been termed "condensational invigoration" (Kogan and Martin, 1994; Seiki and Nakajima, 2014; Saleeby et al., 2015; Sheffield et al., 2015) or "warm-phase invigoration" (Fan et al., 2018). It has recently been suggested that warm-phase invigoration may be more significant and robust than its cold-phase counterpart (Grabowski and Morrison 2016, 2020; Marinescu et al. 2021; Igel and van den Heever 2021).*

*Lines 81–87\*: While convective invigoration hypotheses have proposed stronger updrafts and/or heavier precipitation for convective clouds developing within enhanced aerosol loading conditions, a number of studies have questioned the robustness of these hypotheses for various reasons (Grabowski and Morrison 2016, 2020), one of which is the modulation of these effects by the cloud*

*environment. For example, wind shear (Fan et al., 2009; Lebo and Morrison, 2014; Marinescu et al., 2017), CAPE (Lee et al., 2008; Storer et al., 2010; Storer et al. 2014), boundary layer instability (Marinescu et al., 2021), moisture (Khain et al., 2005, 2008; Tao et al., 2007; Grant and van den Heever, 2015), and aerosol-meteorology covariations (Varble, 2018) have all been found to modulate aerosol impacts on cloud systems.*

3. Line 86-87: Recent studies by Grabowski and Morrison (2016, 2020) that preceded Marinescu et al. (2021) and Igel and van den Heever (2021) should be cited here as suggesting that warm phase invigoration is well founded but cold phase invigoration is not.

   Agreed, we have changed Line 86-87 as follows:

   *Lines 78–80\*: It has recently been suggested that warm-phase invigoration may be more significant and robust than its cold-phase counterpart (Grabowski and Morrison 2016, 2020; Marinescu et al. 2021; Igel and van den Heever 2021).*

4. Is inversion layer strength the lower tropospheric stability, the estimated inversion strength, or some other metric? Can that be clarified?

   We have now clarified the parameter description with relevant references in Table 2.

**Table 2. The ten environmental parameters perturbed in this study and their range.**

| Parameter Description | | Parameter Range | References |
|---|---|---|---|
| Boundary layer | Potential temperature: constant with height within the boundary layer | [285, 300] K | Crosman and Horel (2010) |
| | Relative humidity: constant with height within the boundary layer | [75, 95] % | |
| | Height: distance from the surface to the top of boundary layer | [100, 1000] m | |
| Inversion layer | Strength: lapse rate of the potential temperature, starting immediately at the top of the boundary layer | [1, 15] K km$^{-1}$ | |
| | Depth: distance from the top of boundary layer to the top of inversion layer | [100, 1000] m | |
| Wind | Zonal wind speed: constant with height | [−5, 5] m s$^{-1}$ | |
| Sea surface | Temperature difference between the sea surface and the lowest model level atmosphere (i.e., boundary layer potential temperature) | [−10, 10] K | Igel et al. (2018) |
| | Horizontal gradient of sea surface temperature: linearly applied from the coastline to further offshore | [−0.02, 0.02] K km$^{-1}$ | Reynolds et al. (2007) |

| Land surface | Temperature difference between the land surface and the lowest model level atmosphere | [0, 10] K | Igel et al. (2018) |
|---|---|---|---|
| | Soil saturation fraction: constant over the 11 soil levels with saturation volumetric moisture content of 0.420 m$^3$ m$^{-3}$ (i.e., saturation fraction = 1.0) | [0.1, 0.9] | Rodell et al. (2004) |

5. Line 269: What "instability" is being referred to here? If it is CAPE, then it can be influenced by mixed layer depth but isn't necessarily and certainly isn't monotonically related.

   This has been reworded based on the removal of the ingredient terminology and is no longer relevant.

6. Line 272: Is the implication here that there is a separate lifting mechanism beyond convergence along the sea breeze front? If so, please clarify.

   No that was not our intended implication. We have clarified this sentence.

   *Lines 266–267\*: We now turn our attention to the vertical lift provided by the convergence along the sea breeze front in all of the simulations.*

7. Lines 351-352: What is the difference between "convective instability" for sea breeze initiated convection and "thermal buoyancy" for convection ahead of the sea breeze? Also, are you sure that convective clouds ahead of the sea breeze are in fact always buoyant and instead not at times just negatively buoyant saturated tops of boundary layer dry thermals decelerating in a stable layer? Figure 8 shows that most shallow cloud situations have no mixed layer CAPE.

   The reviewer makes a good point. We meant to separate two sources of instability to convection here: boundary layer instability achieved through surface heating and atmospheric columnar instability such as CAPE since this is more representative of deep convection, as the reviewer noted. Also, we have clarified the following statement as most simulations have zero to very low mixed-layer CAPE (Figure 5e).

   *Lines 342–345\*: We examine these two types of convection separately as they are driven by different processes, with sea breeze convergence and convective instability being critical to the sea breeze-initiated convection, and daytime surface heating and boundary layer mixing being important to the daytime cumulus developing ahead of the sea breeze.*

8. Line 522: Missing "conditions" at end of sentence.

   Done.

---

## Author Response (AR2)

**Editor:**

As you can see the reviewers, who wrote two excellent commentaries, are now largely satisfied with your manuscript although reviewer 2 has some fairly minor clarifications they would like to see addressed. Myself I feel the manuscript is quite interesting and laudable for its emphasis on the radiative effects on the environmental conditions affecting convective cloud formation rather than any "indirect effect" on e.g. convective invigoration. A minor suggestion I have is to emphasize this aspect in the title, to make it more clear that it is the **radiative perturbation that is being considered.**

Thank you very much for your positive comments on our manuscript. We also appreciate your thoughts regarding the title. We agree with your suggestion and have changed the title to **"The Modulation of Aerosol Direct and Indirect Effects on Tropical Sea Breeze Systems by Low-Level Thermodynamic and Surface Conditions"** to better reflect the radiative forcing. We have kept the word "Indirect" in the title too given that we do examine the modulation of some of these processes too.

**Reviewer #2:**

I appreciate the authors' consideration of my comments and their thorough revisions and responses. Most of my concerns have been addressed, and I just have some minor revisions involving clarifying computations, figures, and discussions to consider below.

The authors would like to thank Reviewer 2 for their additional helpful and constructive suggestions. We have addressed each of their comments below. The reviewer comments are in black font, while the authors' responses are in red for contrast.

1.  I'm not entirely clear on how the maximum sea breeze updraft is being computed. It is within 1 km of the surface front but is it anywhere in the column or is it confined to the low levels where the front is forcing air upward? It seems like this should be confined to the lowest levels to assess the vertical motion directly associated with the sea breeze induced convergence (as opposed to buoyantly induced updrafts) with cloud base vertical wind speed being perhaps most relevant. If that is the case, please state that or explain why the calculation used is appropriate for analyzing the sea breeze induced vertical motion.

    In Figures 5i and j, the maximum sea breeze updraft was computed within 1 km of the surface front (either ahead of or behind the front) throughout the depth of the column. The reviewer does, however, make a good point. Following the reviewer's suggestion, we updated Figure 5g by presenting the maximum updraft velocity assessed within 1 km ahead

of and behind the identified surface-based sea breeze front and below 1 km AGL. Most of the cloud bases fall around 1 km AGL and so this seemed to be appropriate. That said, we also tested a 2 km AGL threshold and the results remained similar.

Please note that the maximum updraft discussed in Section 5.2 is different from that shown in Figures 5g and 5h, in that in Section 5.2 we are looking at the maximum values over the entire land and throughout all vertical levels.

We have added the following sentence to address this concern:

> Lines 273–276: We assess the low-level (below 1 km AGL) maximum vertical velocity within $\pm$ 1 km of the surface location of the objectively identified front in order to account for any forward bulging or backward tilting of the frontal boundary in relation to the identified location of the front at the surface, as well as to ensure that the updrafts are primarily driven by the frontal convergence as opposed to the possibility of buoyant forcing.

[Figure]

**Figure 5. Histograms of (a) the land-surface latent heat flux (W m⁻²), (c) the surface-based mixed layer depth (km), and (e) the logarithm of the mixed-layer CAPE (J kg⁻¹), all of which are averaged for each ensemble from the western domain edge to 50 km ahead of the algorithm-identified sea breeze front between 1200–1800 LT; (g) the maximum inland extent of the sea breeze front (km); and (i) the maximum low-level (below 1 km AGL) updraft velocities within ±1 km of the algorithm-identified sea breeze front (m s⁻¹). The median values of each characteristic are marked by the red and blue thin vertical lines. The light blue shading and the red lines represent rOn-500 and rOn-2000 ensembles, respectively. Figures (b, d, f, h, and j) are histograms of the differences in the corresponding fields shown in (a, c, e, g, and i) arising due to aerosol loading (rOn-2000 minus rOn-500). The dashed magenta lines indicate where the difference between rOn-2000 and rOn-500 is zero. The bin width of each histogram is marked at the upper corner of each panel.**

2. For Figure 5, the binning can be improved for several panels including e, f, i, and j where most samples are located within only a couple bins. In addition, for the mixed layer CAPE distribution, it would be good to show how many are 0. Similarly, can the bins be made finer in Fig. 6, 7b, and 10 to show more details? I realize there is a limit on the sample size but there is probably a bit more detail that can be communicated. Mean and median differences in the difference panels might also help.

As shown in #1, we have now updated Figure 5e by taking the logarithm of the CAPE values, and we have increased the bin width in Figure 5f. Figures 5i and j were updated by taking the maximum updraft values below 1 km AGL as addressed in comment #1 above. We also changed the bin widths in Figures 6, 7, and 10 to better represent more details of these distributions. We added the following sentence as there is no simulation with averaged CAPE of zero.

Lines 259–260: The minimum CAPE values in rOn-500 and rOn-2000 are 1.8 and 2.0 J kg$^{-1}$.

[Figure]

**Figure 6.** Similar to Figure 5 but for histograms of the (a) low cloud (cloud top height < 4 km) fractional contribution to the total number of cloudy columns (low cloud columns / all cloudy columns) and (c) the maximum cloud top height. The right column (b and d) represents rOn-2000 minus rOn-500 difference histograms for the corresponding fields (a and c) in the left column. The dashed magenta lines indicate where the difference between rOn-2000 and rOn-500 is zero. The median values of each characteristic are marked with vertical lines in the left column. The bin width of each histogram is marked at the upper corner of each panel.

[Figure]

**Figure 7. Similar to Figure 5 but for (a) histograms of the maximum updraft velocity in rOn-500 and rOn-2000 and (b) a histogram of the maximum updraft velocity differences arising from aerosol loading (rOn-2000 minus rOn-500).**

[Figure]

[Figure]

[Figure]

Figure 10. A histogram of the land-averaged accumulated surface precipitation at sunset (1800 LT) in rOn-500 (blue) and in rOn-2000 (red) for simulations that produce at least 0.1 mm of the land-averaged accumulated precipitation in rOn-500. The absolute and percent differences between rOn-500 and rOn-2000 are shown in (b) and (c), respectively, where the percentage differences are with respect to rOn-500. The dashed magenta line in (b) and (c) indicates where the difference between rOn-2000 and rOn-500 is zero. The bin width of the histogram is marked on each panel.

3. Fig 6c shows a greater median max cloud top height for rOn-500 but the bars would imply something different (i.e., the lowest height bin has many more samples from rOn-500 with the second highest bin shows similar numbers). This must imply that the distribution of values in the second bin (2-4 km heights) is quite different between the 2 runs? If so, I wonder if it makes sense to show more bins so that this is clearer. That would decrease values in the low sample bins, but the peak value would then decrease so you could decrease the y-axis and/or you could also apply non-linear bin widths.

   Thank you for this suggestion. We have changed the bin width of Figure 6c to 0.5 km to better present the distribution of cloud top height from 2 to 4 km more clearly, as shown in #2. As the reviewer suspected, this does better demonstrate the link between the median values and the bin distribution.

4. Is it appropriate to call all the updrafts "convective", e.g., in the Section 5.2 title? The sea breeze will certainly force updrafts that are not convecting (i.e., not buoyant). Perhaps these won't be reflected in the max updraft values being analyzed but that isn't completely clear either.

Thank you for raising this question. We looked into this further. We now make sure that the maximum updraft velocities are indeed convective (i.e., buoyant) by taking the maximum of the updrafts which are greater than 1 m s$^{-1}$ AND have a net positive instantaneous vertical acceleration contribution from the sum of the thermal buoyancy and the condensate loading terms. Adding this constraint on the buoyancy did not qualitatively change the maximum convective updraft results presented in Figures 7 and 8. We updated Figures 7 and 8 to present "convective" updrafts accordingly. We also added the following sentence to the manuscript:

Lines 336–342: To ensure that the maximum updrafts are convective (i.e., positively buoyant) and represent the intensity of sea breeze-initiated convection, we take the maximum updrafts with a net positive instantaneous vertical acceleration contribution from the sum of the thermal buoyancy ($g\frac{\theta'}{\theta_0} + gr'_v\frac{(1-\varepsilon)}{\varepsilon}$) and condensate loading ($-gr_c$) terms in the following vertical momentum equation: $\frac{\partial w}{\partial t} \approx g\frac{\theta'}{\theta_0} + gr'_v\frac{(1-\varepsilon)}{\varepsilon} - gr_c - \frac{1}{\rho}\frac{\partial p'}{\partial z} - w\frac{\partial w}{\partial z} - u\frac{\partial w}{\partial x} - v\frac{\partial w}{\partial y}$ where $w$ is vertical velocity, $g$ is the gravitational acceleration of 9.8 m s$^{-1}$, $\theta'$ is the perturbation potential temperature, $\theta_0$ is the base-state potential temperature, $r'_v$ is the perturbation water vapor mixing ratio, $\varepsilon$ is the ratio of dry air to water vapor gas constants, $r_c$ is the total condensate mixing ratio.

5.  In discussing Fig. 8, technically rOn-2000 has peak updrafts > 10 m/s with boundary layer potential temperature < 299 K even though the text states that it doesn't have updrafts > 10 m/s until 299 K. Please clarify this discussion. In addition, it isn't explained why this is the case. There is discussion of a reduction in boundary layer temperature with enhanced aerosol loading but for a given temperature, deep convection initiation is also reduced with enhanced aerosol loading. Is this related to a change in the capping inversion strength or something else?

Thank you for your comment. We clarified Figure 8 by marking the deep convection simulations in black-lined circles. The requirement on deep convective updrafts is that they have cloud top heights greater than 7 km AGL. We also clarified the following sentences.

Lines 363–366: It is clear from this figure that the sea breeze-initiated deep convective mode (maximum cloud top height greater than 7 km) updrafts in rOn-500 occur in all of the ensemble members in which the initial boundary layer potential temperature is 297 K or greater, and in which the mixed layer CAPE is greatest (not shown).

Lines 367–371: For instance, in rOn-500 while 4 of 12 deep convective simulations with the maximum updraft velocity greater than 10 m s$^{-1}$ have initial boundary layer potential temperatures between 297 and 299 K (Figure 8a), the only simulations in rOn-2000 with the maximum cloud top height greater than 7 km and the maximum updraft velocity greater than 10 m s$^{-1}$ occur for initial boundary layer potential temperature greater than or equal to 299 K (Figure 8b).

Lines 375–378: Also, if we consider those simulations with the same initial boundary layer potential temperatures, the presence of aerosol reduces deep convection initiation. This is due to the fact that aerosol loading leads to less deep convection initiation due to reduced surface temperatures through the scattering of surface downwelling shortwave radiation by aerosol particles.

[Figure]

**Figure 8. Pairwise scatterplots for the maximum updraft velocity (m s⁻¹) versus the initial boundary layer potential temperature (K) for the (a) rOn-500 and (b) rOn-2000 ensembles. The vertical dashed grey lines refer to the potential temperature thresholds described in the text. Simulations with deep convection, identified by the maximum cloud top height > 7 km, are marked with black circles.**

6. It could be useful to know what the percentage decrease in precipitation is in addition to the absolute change in Figure 10 since the meaningfulness of the absolute changes is unclear.

We have updated Figure 10 to include the percentage decrease in precipitation, as shown in #2.

7. In Figure 11, I'm confused by the double peaks in the liquid-ice phase change rates in panels c-e. What is causing the lowest level peaks between 1 and 2 km? Isn't the temperature always well above freezing at these heights? Is this related to hail in some way?

The location of different peaks in Figures 11c–e, particularly for those between 1 and 2 km, are due to different initial conditions. We mentioned this in Lines 467–471. Note that the 1200–1800 LT averaged freezing level ranges from 1.64 to 4.66 km in rOn-500 for the 36 precipitating simulations. In our perturbed parameter ensemble simulations, boundary layer potential temperature ranges from 285 K to 300 K. Depending on other parameter setups in the initial conditions, in some cases, the 0 ºC level is located between 1 and 2 km. For example, two of 36 precipitating pairs have a freezing level (averaged over the land domain and during the afternoon) of 1.64 and 2.12 km in rOn-500, respectively. These lower freezing levels are primarily due to their initial boundary layer potential temperature being at the lower end of the range.

Lines 470–472: It should also be noted that the freezing level varies from simulation to simulation due to the different initial temperature and moisture profiles. For example, two of 36 precipitating pairs have a freezing level (averaged over the land domain and during the afternoon) of 1.64 and 2.12 km in rOn-500, respectively. These lower freezing levels are primarily due to their initial boundary layer potential temperature being at the lower end of the range being tested.

The additional figure below shows the rOn-2000 minus rOn-500 melting of hail process rates, averaged over land between 1200 and 1800 LT. The two pairs with a low freezing level are marked with green lines to answer the reviewer's question regarding hail.

[Figure]

[Figure]

**Figure 11. Aerosol-induced differences to the processes generating rain for the range of environmental conditions tested in these large ensemble experiments. Shown are the rOn-2000 minus rOn500 differences (a) cloud to rain; (b) rain to vapor; (c) melting of ice; (d) rain to ice; and (e) ice to rain rates (see the text for an explanation of these processes) using the 36 ensemble pairs that produce at least 0.1 mm of land-averaged surface accumulated precipitation in rOn-500. Process rates are averaged over the land domain between 1200 and 1800 LT. (f) The rOn-2000 minus rOn-500 raindrop diameter averaged over the land between 1200 and 1800 LT. The thin black lines in all of the figures are from all 36 pairs, and the thick grey lines in a and b are the means of the rOn-2000 minus rOn-500 for the 36 pairs. The dashed magenta lines indicate where the difference between rOn-2000 and rOn-500 is zero.**

8.  In the discussion of more evaporation in rOn-500 relative to rOn-2000 in Fig. 11, the reason provided is that there are more numerous smaller raindrops in rOn-500, but these rates are averaged across all domain grid points over land, so how is it known that drop size and number differences are the cause as compared to just more widespread or heavier rain rates (as panel a implies), which would also increase evaporation?

    We added a panel in Figure 11, showing the rOn-2000 minus rOn-500 averaged rain drop diameter (Figure 11f). Figure 11f shows a robust increase in raindrop diameter in rOn2000 due to enhanced aerosol loading across 36 pairs of precipitating simulations, supporting lines 474–477.

    Lines 474–477: Similarly, average rain evaporation rates (i.e., rain-to-vapor) shown in Figure 11b are also greater in magnitude in rOn-500 than rOn-2000, demonstrating that the population of less numerous but larger raindrops formed in rOn-2000 (Figure 11f) evaporate less readily. The production of populations of fewer but larger raindrops in polluted conditions has been observed previously (e.g., Altaratz et al. 2007; Storer and van den Heever 2013).

9.  Line 490, "loading, that surface" should be "loading, the surface".

    Done.